# Bayesian Optimization with Cost-varying Variable Subsets

**Sebastian Shenghong Tay[1][2], Chuan Sheng Foo[2][3], Daisuke Urano[4],**
**Richalynn Chiu Xian Leong[4], Bryan Kian Hsiang Low[1]**
[1]Department of Computer Science, National University of Singapore
[2]Institute for Infocomm Research (I2R), A*STAR, Singapore
[3]Centre for Frontier AI Research (CFAR), A*STAR, Singapore
[4]Temasek Life Sciences Laboratory, Singapore
`sebastian.tay@u.nus.edu`, `foo_chuan_sheng@i2r.a-star.edu.sg`,
`daisuke@tll.org.sg`, `richalynn@tll.org.sg`, `lowkh@comp.nus.edu.sg`

## Abstract

We introduce the problem of *Bayesian optimization with cost-varying variable subsets* (BOCVS) where in each iteration, the learner chooses a subset of query variables and specifies their values while the rest are randomly sampled. Each chosen subset has an associated cost. This presents the learner with the novel challenge of balancing between choosing more informative subsets for more directed learning versus leaving some variables to be randomly sampled to reduce incurred costs. This paper presents a novel Gaussian process upper confidence bound-based algorithm for solving the BOCVS problem that is provably no-regret. We analyze how the availability of cheaper control sets helps in exploration and reduces overall regret. We empirically show that our proposed algorithm can find significantly better solutions than comparable baselines with the same budget.

## 1 Introduction

*Bayesian optimization* (BO) is a powerful framework for the sample-efficient optimization of costly-to-evaluate black-box objective functions [11] and has been successfully applied to many experimental design problems of significance such as hyperparameter optimization [6, 39], chemical synthesis [30], and particle accelerator control [29], among others. Conventional BO assumes that the learner has full control over all query variables (i.e., all variables in the input to the objective function). However, in many real-world optimization problems, some of the query variables may be subject to randomness affecting their values. In some cases, the randomness affecting a specific variable can be eliminated (by allowing the learner to select its value), but at a cost. We illustrate with a few concrete scenarios: In precision agriculture, consider a farm aiming to find the optimal conditions for largest crop yield where the query variables are a set of soil nutrient concentrations (e.g., Ca, B, $NH_3$, K) and pH. The farm may rely on the naturally-occurring quantities of these nutrients in the available soil, but these quantities will be randomly sampled. Alternatively, they may control some subset of these quantities (via manufactured soil and fertilizers) at a higher cost. In advanced manufacturing where random variation occurs in every operation [34], certain specifications of a product may be left unspecified by the manufacturer and randomly determined, or specified but at a higher cost. In ad revenue maximization or crowdsourcing where information is gathered from a large number of individuals via ad platforms or crowdsourcing platforms such as Amazon Mechanical Turk, suppose that the query variables describe the demographics of the individual, such as country of origin or income level. The learner may allow the platform to randomly assign the task to any individuals, or the learner may demand a specific subgroup of individuals at a higher cost. In all these practical scenarios, the goal is to find the maximizer with as little incurred cost as possible. At each query iteration, the learner is

faced with the non-trivial problem of deciding which variables to specify (for more directed learning) vs. which variables to allow to be randomly sampled (to reduce incurred costs), in addition to the usual BO problem of deciding the specified variables' values.

To the best of our knowledge, there are no existing works that tackle this problem precisely. The work of Hayashi et al. [13] introduced the problem of *BO with partially specified queries* (BOPSQ) in which the subset of deterministically selected variables (*control set*) and randomly sampled variables (*random set*) can also be chosen by the learner, but it does not consider the costs incurred by such choices. This is a non-trivial limitation as the presence of costs can significantly alter the learner's decisions. Under such a formulation, if a control set is a strict subset of another, then the former will never be chosen as there is no benefit to having variable values be randomly sampled instead of chosen by the learner. Consequently, if there exists a control set that includes all the variables in a query, then all other control sets will not be used and the problem reduces to conventional BO. In practice, however, the availability of other control sets confers an advantage if these other control sets are cheaper. Having access to cheaper but more random control sets allows the learner to explore the query space cheaply and then use costlier but more deterministic control sets to exploit high-value regions. BOPSQ in its current formulation excludes the analysis of such strategies and is akin to multi-fidelity BO [15] but without modeling the costs of the different information sources: In this case, the learner would simply choose the highest-fidelity information source all the time, thus making the problem setting trivial.

This paper introduces the problem of *BO with cost-varying variable subsets* (BOCVS) that explicitly models the cost of each control set and is more useful in practical scenarios. Our work generalizes BOPSQ and argues that BOCVS problems are much richer when analyzed from a similar perspective as multi-fidelity BO, and the various control sets are treated as information sources with different levels of usefulness and costs. By using cheap control sets for exploration and expensive control sets for exploitation, we show that with an appropriately designed algorithm, a learner can find significantly better solutions with a lower cost expenditure. To achieve this, we leverage the *Gaussian process upper confidence bound* (GP-UCB) acquisition function [7, 32] to design a novel *no-regret* algorithm, i.e., its incurred simple regret tends to $0$ as the number of iterations tends to infinity, and the algorithm's best chosen query converges to the optimal solution. We additionally analyze the impact of the availability of cheaper control sets on the regret incurred by the most expensive control set. We observe that our algorithm generally outperforms the non-cost-aware baselines, while simple extensions based on Thompson sampling, maximizing UCB or expected improvement-based acquisition scores per unit cost [31, Sec. 3.2] either fail to converge or fail to utilize cheap control sets effectively. Concretely, the contributions of our work in this paper include the following:

- We introduce the BOCVS problem (Sec. 4) and solve it by designing a novel UCB-based algorithm (Sec. 4.1) with a theoretical analysis of its properties, including the conditions under which it is provably no-regret and the impact of the availability of cheaper control sets on the regret incurred by the most expensive control set, and discuss the practical considerations (Sec. 4.2);
- We empirically evaluate the performance of our proposed algorithm against the baselines under several experimental settings with synthetic and real-world datasets (Sec. 5), including a plant growth dataset and an airfoil self-noise dataset corresponding, respectively, to the precision agriculture and advanced manufacturing use cases motivated earlier in this section.

## 2 Related Work

The work of Hayashi et al. [13] introduced *BO with partially specified queries* (BOPSQ) and tackled the problem with Thompson sampling. However, it fails to consider the relative costs of control sets, which hinders the learner's ability to take advantage of all control sets even in the presence of more deterministic control sets. The work of Oliveira et al. [25] proposed BO with uncertain inputs in which the executed query is sampled from a probability distribution depending on the proposed query. Though related, its problem setting is motivated more by uncertainty in the input query even post-observation and does not involve variable subset selection. These two works are part of a line of research investigating BO in situations where the learner may not have full control over all variables in a query, which includes BO for expected values [38], risk-averse BO [5, 21, 22], and distributionally robust BO [17, 24, 37]. These works also do not consider variable subset selection. Our treatment of the BOCVS problem is inspired by multi-fidelity BO in which the learner has access to cheap, low-fidelity surrogates of the true objective function [15, 27, 35, 36]. In such works (and in

ours), modeling costs is crucial as the learner would simply choose the highest-fidelity information source (in ours, the maximally deterministic control set) otherwise. While the general idea of paying less for potentially less informative queries is similar, our problem setting is fundamentally different: The lack of informativeness comes from the uncertainty of the executed query as opposed to a bias in the observed function values.

The BOCVS setting may be viewed as a special case of causal BO as formulated by Aglietti et al. [1] and continued in several works [2, 4]. Specifically, our setting is a case in which there are no 'non-manipulative' variables and the causal DAG is such that all input variables have no parents and are parents of the output variable. Nevertheless, we believe our focus on this special case has value as it allows us to derive useful theoretical results such as algorithm regret bounds that, to the best of our knowledge, do not exist for the completely general causal BO setting at the time of writing. The work of Sussex et al. [33] includes a regret bound, but is also a special case of [1], and has little overlap with our work as it does not consider costs of control sets or explicit probability distributions over input variables. We believe that our work is sufficiently general to be useful for practical scenarios (where the full causal BO apparatus may be unnecessary), and is also a stepping stone towards theory for the general case.

## 3 BO and Gaussian Processes

We will first give a brief review of conventional BO [11]. Given a query set $\mathcal{X}$ and an objective function $f : \mathcal{X} \to \mathbb{R}$, a learner wishes to find the maximizing query $\mathbf{x}^* := \operatorname{argmax}_{\mathbf{x} \in \mathcal{X}} f(\mathbf{x})$. However, $f$ is black-box (i.e., not available in closed form) and can only be learned by submitting a query $\mathbf{x}_t \in \mathcal{X}$ in each iteration $t$ for function evaluation and receiving a noisy observation $y_t := f(\mathbf{x}_t) + \xi_t$ where each $\xi_t$ is i.i.d. $\sigma$-sub-Gaussian noise with zero mean. Each function evaluation is assumed to be expensive in some way, such as in terms of money or time spent. So, the learner must be sample-efficient and find $\mathbf{x}^*$ in as few iterations as possible. BO achieves sample efficiency by leveraging a Bayesian model to represent a probabilistic belief of the function values at unobserved regions of $\mathcal{X}$ in a principled manner. While any Bayesian model may be used for BO, *Gaussian processes* (GPs) [42] are a common choice as they enable exact posterior inference: The GP posterior belief of $f$ at any query $\mathbf{x} \in \mathcal{X}$ after $t$ iterations is a Gaussian with posterior mean and variance given by

$$\mu_t(\mathbf{x}) := \mathbf{k}_t(\mathbf{x})^\top (\mathbf{K}_t + \lambda \mathbf{I})^{-1} \mathbf{y}_t , \quad \sigma_t^2(\mathbf{x}) := k(\mathbf{x}, \mathbf{x}) - \mathbf{k}_t(\mathbf{x})^\top (\mathbf{K}_t + \lambda \mathbf{I})^{-1} \mathbf{k}_t(\mathbf{x}) \quad (1)$$

where $\mathbf{y}_t := (y_j)_{j=1}^t \in \mathbb{R}^t$, $k$ is a positive semidefinite *kernel* (covariance function), $\mathbf{k}_t(\mathbf{x}) := (k(\mathbf{x}, \mathbf{x}_j))_{j=1}^t \in \mathbb{R}^t$, $\mathbf{K}_t := (k(\mathbf{x}_j, \mathbf{x}_{j'}))_{j,j'=1}^t \in \mathbb{R}^{t \times t}$, and $\lambda$ is an algorithm parameter; if the noise is a Gaussian with variance $\sigma^2$, then the true posterior is recovered with $\lambda = \sigma^2$. The kernel $k$ is an important modeling choice as the GP posterior mean will reside in the *reproducing kernel Hilbert space* (RKHS) associated with $k$. For simplicity, we assume w.l.o.g. that $k(\mathbf{x}, \mathbf{x}') \leq 1$ for any pair of queries $\mathbf{x}, \mathbf{x}' \in \mathcal{X}$. Kernel $k$ affects the *maximum information gain* (MIG) defined as

$$\gamma_T(\mathcal{X}) := \max_{\{\mathbf{x}_t\}_{t=1}^T \subseteq \mathcal{X}} 0.5 \log \left| \mathbf{I} + \lambda^{-1} \mathbf{K}_T \right| .$$

The MIG characterizes the statistical complexity of a problem and plays an integral role in the theoretical analysis. For the commonly used squared exponential kernel, $\gamma_T(\mathcal{X}) = \mathcal{O}((\log T)^{d+1})$, while for the Matérn kernel with $\nu > 1$, $\gamma_T(\mathcal{X}) = \mathcal{O}(T^{d(d+1)/(2v+d(d+1))}(\log T))$ [32]. Importantly, $\gamma_T(\mathcal{X})$ is increasing in the volume of $\mathcal{X}$ [32, Theorem 8].

## 4 BO with Cost-varying Variable Subsets (BOCVS)

The BOCVS problem consists of a compact query set $\mathcal{X} \subset \mathbb{R}^d$ and an objective function $f : \mathcal{X} \to \mathbb{R}$ in the RKHS of $k$ with the RKHS norm upper bounded by $B$. For simplicity, assume w.l.o.g. that $\mathcal{X} = [0, 1]^d$. Let $[d] := \{1, 2, ..., d\}$. The learner is given a collection $\mathcal{I} \subseteq 2^{[d]}$ of *control sets* indexed by $1, 2, \ldots, m := |\mathcal{I}|$. Each control set $i \in [m]$, denoted by $\mathcal{I}_i \subseteq [d]$, indicates the variables in a query with values that can be chosen by the learner. The complement $\bar{\mathcal{I}}_i := [d] \setminus \mathcal{I}_i$ of $\mathcal{I}_i$ is the corresponding *random set* indicating the variables in a query with values that will be randomly sampled from some distribution. A query $\mathbf{x} \in \mathcal{X}$ can be represented by a combination of *partial queries* $[\mathbf{x}^i, \mathbf{x}^{-i}]$ comprising the *control partial query* $\mathbf{x}^i := (x_\ell)_{\ell \in \mathcal{I}_i}$ (i.e., $\mathbf{x}^i$ collects the variables

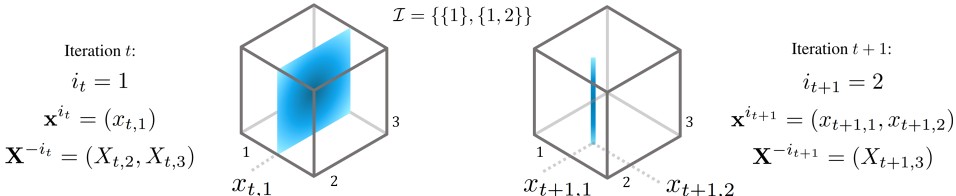

Figure 1: Two iterations in a BOCVS problem setting. The grey boxes are isometric views of a query set $\mathcal{X} \subset \mathbb{R}^3$. The blue regions depict the probability densities of random vectors $[\mathbf{x}^{i_t}, \mathbf{X}^{-i_t}]$ and $[\mathbf{x}^{i_{t+1}}, \mathbf{X}^{-i_{t+1}}]$. In iteration $t$, the learner chooses the control set $i_t = 1$ and specifies the value (of the first variable $x_{t,1}$) in control partial query $\mathbf{x}^{i_t}$, while the last two variables $X_{t,2}, X_{t,3}$ in random partial query $\mathbf{X}^{-i_t}$ will be randomly sampled. In iteration $t + 1$, the learner chooses the control set $i_{t+1} = 2$ and specifies the values (of the first two variables $x_{t,1}, x_{t,2}$) in control partial query $\mathbf{x}^{i_{t+1}}$, while the last variable $X_{t,3}$ in random partial query $\mathbf{X}^{-i_{t+1}}$ will be randomly sampled.

indexed by $\mathcal{I}_i$) and the *random partial query* $\mathbf{x}^{-i} \coloneqq (x_\ell)_{\ell \in \overline{\mathcal{I}}_i}$ where $x_\ell$ denotes the $\ell$-th variable in the query vector $\mathbf{x}$. Note that $[\mathbf{x}^i, \mathbf{x}^{-i}]$ is not a simple vector concatenation as the variables may need to be reordered according to their indices. Furthermore, let $\mathcal{X}^i \coloneqq \{\mathbf{x}^i \mid \mathbf{x} \in \mathcal{X}\}$.

In iteration $t$, the learner chooses control set $i_t \in \mathcal{I}$ and specifies the values in control partial query $\mathbf{x}^{i_t}$. The random partial query $\mathbf{x}^{-i_t}$ will then be randomly sampled from the environment. For example, if $d = 4$ and $\mathcal{I}_{i_t} = \{1, 3\}$, then $\overline{\mathcal{I}}_{i_t} = \{2, 4\}$ and the learner will be able to choose the values in $\mathbf{x}^{i_t}$ (i.e., the 1$^{\text{st}}$ and 3$^{\text{rd}}$ variables) but not those in $\mathbf{x}^{-i_t}$ (i.e., the 2$^{\text{nd}}$ and 4$^{\text{th}}$ variables). The full query in iteration $t$ is then $\mathbf{x}_t = [\mathbf{x}^{i_t}, \mathbf{x}^{-i_t}] = (x_{t,\ell})_{\ell \in [d]}$. Each observed variable $x_{t,\ell}$ for $\ell \in \overline{\mathcal{I}}_{i_t}$ is a realization of a random variable $X_{t,\ell} \sim \mathcal{P}_\ell$. The observed $\mathbf{x}^{-i_t}$ is then a realization of the random vector $\mathbf{X}^{-i_t} \coloneqq (X_{t,\ell})_{\ell \in \overline{\mathcal{I}}_{i_t}} \sim \mathbb{P}^{-i_t}$ where $\mathbb{P}^{-i_t}$ is the product measure $\bigtimes_{\ell \in \overline{\mathcal{I}}_{i_t}} \mathcal{P}_\ell$. In other words, each variable in a random partial query is independently sampled from a probability distribution that governs that variable. All distributions are assumed to be known. The learner then observes $y_t \coloneqq f(\mathbf{x}_t) + \xi_t$ where each $\xi_t$ is i.i.d. $\sigma$-sub-Gaussian noise with a zero mean. Fig. 1 illustrates two iterations in a BOCVS problem setting.

The learner wishes to find the optimal control set $i^*$ and specified values in control partial query $\mathbf{x}^{i^*}$ that maximize the expected value of $f([\mathbf{x}^i, \mathbf{X}^{-i}])$ where the expectation is w.r.t. $\mathbf{X}^{-i} \sim \mathbb{P}^{-i}$:

$$(i^*, \mathbf{x}^{i^*}) \coloneqq \underset{(i, \mathbf{x}^i) \in [m] \times \mathcal{X}^i}{\operatorname{argmax}} \mathbb{E}\big[f([\mathbf{x}^i, \mathbf{X}^{-i}])\big].$$

The learner has an initial budget $C \in \mathbb{R}^+$ and every control set $\mathcal{I}_i$ has an associated cost $c_i > 0$ for all $i \in [m]$. Let the control set indices be defined such that $c_1 \leq c_2 \leq \ldots \leq c_m$.[1]

In every iteration $t$, the learner pays $c_{i_t}$. The learning procedure ends after $T$ iterations when $C - \sum_{t=1}^T c_{i_t} < c_{i_{T+1}}$, i.e., the learner has not enough budget left to pay for the chosen control set. $T$ will now be a random variable depending on the algorithm and the random outcomes of the learning procedure. The cost-varying cumulative regret is defined as

$$R_T \coloneqq \sum_{t=1}^T c_{i_t} \left( \mathbb{E}\big[f([\mathbf{x}^{i^*}, \mathbf{X}^{-i^*}])\big] - \mathbb{E}\big[f([\mathbf{x}^{i_t}, \mathbf{X}^{-i_t}])\big] \right).$$

The regret incurred by choosing a sub-optimal control set and specifying sub-optimal values in the control partial query is weighted by the cost of that control set. This naturally incorporates the notion that the penalty for sub-optimal plays is lower if the play was cheap, while also penalizing using the entire budget on sub-optimal plays, regardless of whether those plays are cheap or expensive. Intuitively, to minimize the cost-varying regret, a learner would attempt to use the cheap control sets (i.e., low $c_i$, low $\mathbb{E}\big[f([\mathbf{x}^i, \mathbf{X}^{-i}])\big]$) to explore the query space, and use the expensive control sets (i.e., high $c_i$, high $\mathbb{E}\big[f([\mathbf{x}^i, \mathbf{X}^{-i}])\big]$) to exploit control partial queries with high expected function values.[1] When all $c_i = 1$, we recover the BOPSQ problem [13], and $C$ is simply the number of iterations

---

[1] While our problem definition does not require that $c_i \leq c_j \Leftrightarrow \max_{\mathbf{x}^i \in \mathcal{X}^i} \mathbb{E}\big[f([\mathbf{x}^i, \mathbf{X}^{-i}])\big] \leq \max_{\mathbf{x}^j \in \mathcal{X}^j} \mathbb{E}\big[f([\mathbf{x}^j, \mathbf{X}^{-j}])\big]$, one might reasonably expect this to be the case in real-world problems, i.e., "better" control sets cost more to specify. This also implies that $\mathcal{I}_i \subseteq \mathcal{I}_j \Rightarrow c_i \leq c_j$.

---

**Algorithm 1** UCB-CVS

1: **Input:** GP with kernel $k$, budget $C$, control sets $\mathcal{I}$, costs $(c_i)_{i=1}^m$, $\epsilon$-schedule $(\epsilon_t)_{t=1}^\infty$
2: **for** iteration $t = 1$ **to** $\infty$ **do**
3:      $g_t := \max_{(i, \mathbf{x}^i) \in [m] \times \mathcal{X}^i} \mathbb{E}\big[u_{t-1}([\mathbf{x}^i, \mathbf{X}^{-i}])\big]$
4:      $\mathcal{S}_1 := \big\{i \in [m] \mid \max_{\mathbf{x}^i \in \mathcal{X}^i} \mathbb{E}\big[u_{t-1}([\mathbf{x}^i, \mathbf{X}^{-i}])\big] + \epsilon_t \geq g_t\big\}$
5:      $\mathcal{S}_2 := \{i \in \mathcal{S}_1 \mid c_i = \min_{j \in \mathcal{S}_1} c_j\}$
6:      $(i_t, \mathbf{x}^{i_t}) := \mathrm{argmax}_{(i, \mathbf{x}^i) \in \mathcal{S}_2 \times \mathcal{X}^i} \mathbb{E}\big[u_{t-1}([\mathbf{x}^i, \mathbf{X}^{-i}])\big]$
7:      **break if** $C - \sum_{\tau=1}^{t-1} c_{i_\tau} < c_{i_t}$
8:      Observe $\mathbf{x}^{-i_t}$ drawn from $\mathbb{P}^{-i_t}$
9:      Observe $y_t := f(\mathbf{x}_t) + \xi_t$
10:     $\mathcal{D}_t := \{(\mathbf{x}_\tau, y_\tau)\}_{\tau=1}^t$
11: **end for**
12: **return** $\mathcal{D}_t$

---

in the learning trajectory. In fact, BOPSQ reduces to a simpler problem if there exists a *full query control set* that allows the learner to choose the values of all $d$ variables. If $[d] \in \mathcal{I}$, then $\mathcal{I}_{i^*} = [d]$ and $\mathbb{E}\big[f([\mathbf{x}^{i^*}, \mathbf{X}^{-i^*}])\big] = \max_{\mathbf{x} \in \mathcal{X}} f(\mathbf{x})$ since expectations of a function are never greater than the maximum value of the function. In other words, the full query control set is guaranteed to be the optimal control set and the BOPSQ problem reduces to one of conventional BO. In general, under BOPSQ, any control set that is a strict subset of another will never be chosen.

## 4.1 UCB-CVS

Alg. 1 describes our UCB-CVS algorithm for solving the BOCVS problem. In iteration $t$, it uses the GP posterior belief of $f$ to construct an *upper confidence bound* (UCB) $u_{t-1}$ of $f$:

$$u_{t-1}(\mathbf{x}) = \mu_{t-1}(\mathbf{x}) + \beta_t \sigma_{t-1}(\mathbf{x})$$

where the sequence $(\beta_t)_{t \geq 1}$ is an algorithm parameter that controls the tradeoff between exploration and exploitation. UCB-based algorithm design is a classic strategy in the stochastic bandits [19, Ch. 7] and BO literature [7, 32] and makes use of the "*optimism in the face of uncertainty*" (OFU) principle [18]: Queries with a large posterior standard deviation (i.e., high uncertainty) are given high acquisition scores as the function values at those queries may be potentially high. UCB-CVS adapts this strategy by taking the expectation of the UCB as part of the acquisition process. Due to the monotonicity of expectation, if $u_{t-1}$ is an upper bound of $f$ (i.e., $u_{t-1}(\mathbf{x}) \geq f(\mathbf{x})$ for any $\mathbf{x} \in \mathcal{X}$), then $\mathbb{E}\big[u_{t-1}([\mathbf{x}^i, \mathbf{X}^{-i}])\big]$ is also an upper bound of $\mathbb{E}\big[f([\mathbf{x}^i, \mathbf{X}^{-i}])\big]$ for any $i \in [m], \mathbf{x}^i \in \mathcal{X}^i$.

UCB-CVS also takes as input an $\epsilon$-schedule $(\epsilon_t)_{t=1}^\infty$ where $\epsilon_t \geq 0$ for all $t$. To choose the control set in iteration $t$, it first computes $g_t$ which is the expected UCB of the best control set and specified values in the control partial query (Step 3). It then collects every control set $i$ that fulfills the condition $\max_{\mathbf{x}^i \in \mathcal{X}^i} \mathbb{E}\big[u_{t-1}([\mathbf{x}^i, \mathbf{X}^{-i}])\big] + \epsilon_t \geq g_t$ into a set $\mathcal{S}_1$ (Step 4). It further reduces this set $\mathcal{S}_1$ to $\mathcal{S}_2$ by retaining only the control sets with the lowest cost (Step 5). Finally, it chooses the control set from $\mathcal{S}_2$ with the largest expected UCB value (Step 6). Each $\epsilon_t$ thus serves as a relaxation that enables exploration with cheaper control sets. Choosing many $\epsilon_t$ to be large results in many iterations of choosing cheaper control sets; conversely, choosing $\epsilon_t = 0$ for all $t$ ignores all costs.

Our first result upper bounds the cost-varying cumulative regret incurred by UCB-CVS. Define the *feasible set* $\widetilde{\mathcal{X}}_i := \times_{\ell=1}^d [a_\ell^i, b_\ell^i]$ for each control set $i$ such that $a_\ell^i = 0, b_\ell^i = 1$ if $\ell \in \mathcal{I}_i$, and $a_\ell^i = \sup\{a \in [0, 1] \mid F_\ell(a) = 0\}, b_\ell^i = \inf\{b \in [0, 1] \mid F_\ell(b) = 1\}$ otherwise, where $F_\ell$ is the CDF of $X_\ell \sim \mathcal{P}_\ell$. $\widetilde{\mathcal{X}}_i$ is a subset of $\mathcal{X}$ in which any query chosen with control set $i$ must reside. Define $T_i$ as the total number of iterations in which control set $i$ is chosen.

**Theorem 4.1.** *With probability at least $1 - \delta$, UCB-CVS (Alg. 1) incurs a cost-varying cumulative regret bounded by*

$$R_T \leq \mathcal{O}\left(\left(B + \sqrt{\gamma_T(\mathcal{X}) + \log \frac{m+1}{\delta}}\right)\left(\sum_{i=1}^m c_i \left(\sqrt{T_i \gamma_{T_i}(\widetilde{\mathcal{X}}_i)} + \log \frac{m+1}{\delta}\right)\right)\right) + c_m \sum_{t=1}^T \epsilon_t$$

*by setting $\beta_t = B + \sigma\sqrt{2(\gamma_{t-1}(\mathcal{X}) + 1 + \log((m+1)/\delta))}$.*

For any appropriately chosen kernel such that $\gamma_T(\mathcal{X}) < \mathcal{O}(\sqrt{T})$ (e.g., commonly used squared exponential kernel, see Sec. 3) and $\epsilon$-schedule such that $\sum_{t=1}^{T} \epsilon_t$ is sublinear in $T$, the cumulative regret incurred will be sublinear in $T$: $\lim_{T \to \infty} R_T/T = 0$. Since the mean of a sequence is no less than the minimum, and all $c_i > 0$, this further implies the desired no-regret property: $\lim_{T \to \infty} \min_{1 \le t \le T} (\mathbb{E}[f([\mathbf{x}^{i^*}, \mathbf{X}^{-i^*}])] - \mathbb{E}[f([\mathbf{x}^{i_t}, \mathbf{X}^{-i_t}])]) = 0$, i.e., the best control set and specified values in control partial query in the algorithm's choices eventually converge to the optimal solution. The proof of Theorem 4.1 relies on choosing an appropriate sequence of $\beta_t$ such that $u_{t-1}(\mathbf{x}) \ge f(\mathbf{x})$ for any $\mathbf{x} \in \mathcal{X}, t \ge 1$ with high probability [7, Theorem 2]. The cumulative regret is bounded by a sum of expectations of posterior standard deviations, which can then be bounded by a sum of posterior standard deviations plus some additional terms [16, Lemma 3] and in turn bounded in terms of the MIG [7, Lemma 4]. The proofs of all results in this paper are provided in Appendix A.

Since each $\gamma_{T_i}(\widetilde{\mathcal{X}}_i)$ is increasing in the volume of $\widetilde{\mathcal{X}}_i$, Theorem 4.1 states that control sets with smaller feasible sets will incur less regret. If the size of a feasible set is taken to be a reasonable surrogate for the diffuseness of the probability distributions involved, Theorem 4.1 then suggests that control sets with corresponding random sets whose probability distributions are less diffuse will incur less regret.[2] Theorem 4.1 also informs us that one sufficient condition on the $\epsilon$-schedule for the cost-varying regret to be sublinear in $T$ is that $\sum_{t=1}^{T} \epsilon_t$ is sublinear in $T$. Our next proposition provides an alternative condition (neither is more general than the other):

**Proposition 4.2.** *If there exists a $\tilde{\epsilon} > 0$ s.t. for all $i \ne i^*$, $\epsilon_t \le \mathbb{E}[f([\mathbf{x}^{i^*}, \mathbf{X}^{-i^*}])] - \max_{\mathbf{x}^i \in \mathcal{X}^i} \mathbb{E}[f([\mathbf{x}^i, \mathbf{X}^{-i}])] - \tilde{\epsilon}$ eventually (i.e., the inequality holds for all $t \ge q$ for some $q \ge 1$), and $\gamma_T(\mathcal{X}) < \mathcal{O}(\sqrt{T})$, then, with probability at least $1 - \delta$, $\lim_{T \to \infty} T_i/T = 0$ for all $i \ne i^*$ and UCB-CVS incurs a cost-varying cumulative regret that is sublinear in $T$ by setting $\beta_t = B + \sigma \sqrt{2 (\gamma_{t-1}(\mathcal{X}) + 1 + \log((m+1)/\delta))}$.*

The above results have shown that with an appropriately chosen $\epsilon$-schedule, UCB-CVS satisfies the no-regret property. However, ignoring all costs by setting $\epsilon_t = 0$ for all $t$ also achieves no-regret. This begs the question: *In what way does a good $\epsilon$-schedule improve UCB-CVS?* Supposing the most expensive control set is the full query control set, the presence of queries chosen with cheaper control sets should reduce the cost-varying regret incurred by the full query control set by ruling out low function value regions and directing the full queries towards high function value regions. Additionally, it is reasonable to conjecture that the more diffuse each variable's (indexed by $\ell$) probability distribution $\mathcal{P}_\ell$ is, the more the cheaper control sets would explore the query space and thus, the lower the cost-varying regret incurred by the full query control set. To derive such a result, the plan of attack is to relate the variances (i.e., notion of diffuseness) of the probability distributions to the distances between queries chosen with the cheaper control sets, followed by analyzing the effect of these distances and the number of times cheaper control sets were played on the MIG term of the most expensive control set. Our next result relates the distance between pairs of queries chosen with control set $i$ to the variance $\mathbb{V}[X_\ell]$ of every probability distribution $\mathcal{P}_\ell$ for $\ell \in \overline{\mathcal{I}}_i$:

**Lemma 4.3.** *Suppose that for each control set $i$, the random variable $Y_i := \left\| [\mathbf{0}, \mathbf{X}_1^{-i}] - [\mathbf{0}, \mathbf{X}_2^{-i}] \right\|^2$ has a median $M_i$ s.t. $\mathbb{E}[Y_i | Y_i > M_i] \le h_i M_i$ for some $h_i > 0$ where $\mathbf{X}_1^{-i}, \mathbf{X}_2^{-i} \sim \mathbb{P}^{-i}$. With probability at least $1 - \delta$, there will be at least $N_i$ non-overlapping pairs of queries $\mathbf{x}$ and $\mathbf{x}'$ chosen by UCB-CVS (Alg. 1) with control set $i$ s.t. $\|\mathbf{x} - \mathbf{x}'\|^2 \ge M_i$ where*

$$N_i = \left\lfloor (T_i - 1)/4 - \sqrt{(T_i/4) \log(1/\delta)} \right\rfloor \quad and \quad M_i \ge (4/(h_i + 1)) \sum_{\ell \in \overline{\mathcal{I}}_i} \mathbb{V}[X_\ell] . \quad (2)$$

From (2), the higher the variances of the distributions that govern the variables in the random set, the larger the lower bound $M_i$ on the squared distance between at least $N_i$ pairs of queries chosen with control set $i$. As expected, the number $N_i$ of pairs increases with $T_i$ (i.e., the total number of iterations in which control set $i$ is chosen). The assumption on $Y_i$ is mild: As long as $Y_i$ has at least 1 non-zero median, it will hold. The assumption excludes the case in which $\mathcal{P}_\ell$ for all $\ell \in \overline{\mathcal{I}}_i$ are degenerate with all probability mass on a single point. With Lemma 4.3, we now derive an alternative regret bound that depends on the variances of the distributions and the number of plays of cheaper control sets:

---

[2]The feasible set of control set $i$ is defined in a worst-case manner, which may be too conservative to be a good surrogate for diffuseness, especially for concentrated probability distributions with non-zero density everywhere. Nevertheless, it facilitates the worst-case analysis of the regret bounds.

**Theorem 4.4.** *Suppose that the following hold:*

- *Assumption of Lemma 4.3 holds;*
- $k(\mathbf{x}, \mathbf{x}')$ *is an isotropic kernel which only depends on distance between $\mathbf{x}$ & $\mathbf{x}'$ and can be written as $k(\|\mathbf{x} - \mathbf{x}'\|)$;*
- *There exists an iteration $r$ s.t. for all $t \leq r, i_t \leq m - 1$, and for all $t > r, i_t = m$ .*

*Then, with probability at least $1 - \delta$, UCB-CVS (Alg. 1) incurs a cost-varying cumulative regret bounded by*

$$
\begin{aligned}
R_T \leq \mathcal{O}\Bigg( & \left( B + \sqrt{\gamma_T(\mathcal{X}) + \log \frac{2m}{\delta}} \right) \left( c_m \left( \sqrt{T\gamma_T(\mathcal{X})} - \mathcal{L} + \log \frac{2m}{\delta} \right) \right. \\
& \left. + \sum_{i=1}^{m-1} c_i \left( \sqrt{T_i \gamma_{T_i}(\widetilde{\mathcal{X}_i})} + \log \frac{2m}{\delta} \right) \right) \Bigg) + c_m \sum_{t=1}^{T} \epsilon_t \\
\mathcal{L} := \lambda & \left( \sum_{i=1}^{m-1} N_i \log \left( V_i - 2k\left(\sqrt{M_i}\right) - k\left(\sqrt{M_i}\right)^2 \right) + W \right)
\end{aligned}
$$

*by setting $\beta_t = B + \sigma \sqrt{2 \left( \gamma_{t-1}(\mathcal{X}) + 1 + \log((2m)/\delta) \right)}$ where $N_i$ and $M_i$ are previously defined in Lemma 4.3, and $V_i$ and $W$ are residual terms defined in Appendix A.5.*

Theorem 4.4 shows that the MIG term pertaining to the most expensive control set $m$ is reduced by $\mathcal{L}$ which increases as $N_i$ increases, which in turn increases as $T_i$ increases. This suggests that an $\epsilon$-schedule that increases the number of times cheaper control sets are played can reduce the MIG term. $\mathcal{L}$ also increases as $k(\sqrt{M_i})$ decreases. For common kernels such as the squared exponential or Matérn kernel with $\nu > 1$ (which satisfy the second assumption on isotropic kernel), $k(\sqrt{M_i})$ decreases as $M_i$ increases, from which we may conclude that higher variance probability distributions governing each $X_\ell$ lead to a larger $\mathcal{L}$ due to (2) and hence a larger decrease on the MIG term. In cases where $c_m \gg c_i$ for all $i \neq m$, a carefully chosen $\epsilon$-schedule can thus lead to a large decrease in the regret bound via $\mathcal{L}$. The third assumption is (informally) approximately true in practice due to the design of UCB-CVS: If a decreasing $\epsilon$-schedule is used, the algorithm will choose the cheaper but sub-optimal control sets at the start. After $\epsilon_t$ has decreased past a certain value, the algorithm will only choose the optimal (and likely most expensive) control set. The proof sketch upper bounds the sum of posterior standard deviations of queries chosen with control set $m$ with the MIG term minus the sum of posterior standard deviations of queries chosen with all other control sets. This latter sum is then lower bounded by a log determinant of the prior covariance matrix which is then decomposed into a sum of log determinants of pairs of queries. The dependence on the distances between the pairs can be made explicit in this form. Neither Theorems 4.1 nor 4.4 is more general than the other.

## 4.2 Practical Considerations

UCB-CVS is presented with the $\epsilon$-schedule formulation for generality and ease of theoretical analysis. In practice, however, the $\epsilon$-schedule is a hyperparameter that is difficult to interpret and choose. We propose a simple *explore-then-commit* (ETC) variant with which the learner only chooses the number of plays of each *cost group* (i.e., defined as a collection of control sets with the same cost that is not the maximum cost). In each iteration, the algorithm will choose the cost group with the lowest cost and non-zero remaining plays, and then choose the control set within that cost group with the largest expected UCB (similar to Step 6 in Alg. 1). Once all cost groups have zero remaining plays, the algorithm chooses the control set with the largest expected UCB among all control sets. This algorithm is highly interpretable and is equivalent to UCB-CVS with a specific sublinear $\epsilon$-schedule (that cannot be known *a priori*). Furthermore, the learner should choose the number of plays adaptively depending on the cost of each cost group. On computational considerations, UCB-CVS may be computationally expensive if the number $m$ of control sets is large (e.g., if every subset of variables is available as a control set and $m = 2^d$) as each control set requires a maximization of the expected UCB (which can be approximated with Monte Carlo sampling). In such cases, the learner has the option to simply ignore any number of control sets to reduce $m$, as long as $i^*$ is not ignored.

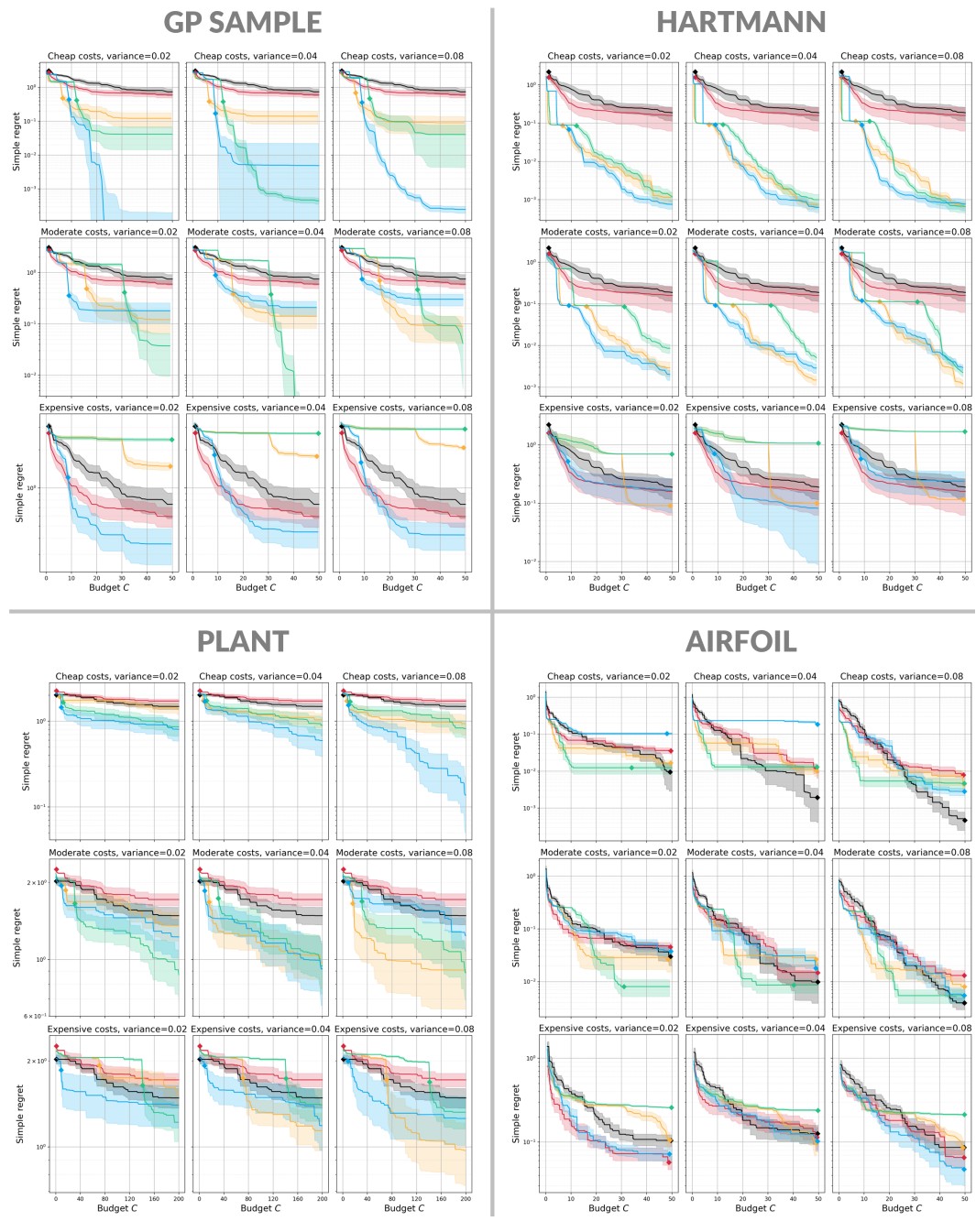

Figure 2: Mean and standard error (over 10 RNG seeds) of the simple regret (lower is better) incurred against cost spent (budget) $C$ by **TS-PSQ**, **UCB-PSQ**, **ETC-50**, **ETC-100**, and **ETC-Ada** with varying objective functions, cost sets, and variances of distributions. A diamond indicates the average budget after which an algorithm only chooses the optimal control set.

# 5 Experiments and Discussion

This section empirically evaluates the performance of the tested algorithms with 4 objective functions: (a) function samples from a GP prior (3-D), (b) the Hartmann synthetic function (3-D), (c) a plant growth simulator built from real-world data where the variables are nutrients such as $NH_3$ and pH (5-D), and (d) a simulator built from the airfoil self-noise dataset (5-D) from the UCI Machine Learning Repository [9]. For the first 2 objective functions, the control sets are all possible subsets of the 3 variables except the empty set, which leads to 7 control sets. For the plant growth objective function, we pick 7 control sets including the full query control set. For the airfoil self-noise objective function, similar to that of [13], we pick 7 control sets of 2 variables each that are not subsets of each other. We use 3 different sets of costs for the 7 control sets: **cheap** ($\{0.01, 0.01, 0.01, 0.1, 0.1, 0.1, 1\}$), **moderate** ($\{0.1, 0.1, 0.1, 0.2, 0.2, 0.2, 1\}$), and **expensive** ($\{0.6, 0.6, 0.6, 0.8, 0.8, 0.8, 1\}$). Using these sets of costs, the control sets are ordered such that $c_i < c_j \Rightarrow \max_{\mathbf{x}^i \in \mathcal{X}^i} \mathbb{E}\big[f([\mathbf{x}^i, \mathbf{X}^{-i}])\big] \leq \max_{\mathbf{x}^j \in \mathcal{X}^j} \mathbb{E}\big[f([\mathbf{x}^j, \mathbf{X}^{-j}])\big]$. These cost sets have fixed the optimal (i.e., last) control set to have a cost of 1. While these cost sets may (at first glance) seem arbitrary, it is the algorithms' *relative performance across these cost sets* rather than the absolute performance on a single cost set that allows us to understand the conditions under which particular algorithms perform better or worse. Real-world applications (unlike the experiments conducted here) will come with their own cost sets defined by real-world constraints. If the real costs can also be categorized in a similar relative way like the above cheap, moderate, and expensive cost sets, then the results are expected to be similar. Every probability distribution $\mathcal{P}_\ell$ is a truncated normal distribution with mean 0.5 and the same variance which is one of 0.02, 0.04, and 0.08 (the uniform distribution on $[0, 1]$ has variance $1/12$).

We compare the performance of our algorithm against that of the baseline Thompson sampling (**TS-PSQ**) algorithm developed in [13]. We test **UCB-PSQ** ($\epsilon$-schedule with $\epsilon_t = 0$ for all $t$) along with the ETC variant of UCB-CVS (Sec. 4.2) with 3 sets of hyperparameters: 50 plays per cost group (**ETC-50**), 100 plays per cost group (**ETC-100**), and a cost-adaptive version with $4/c_j$ plays per cost group where $c_j$ is the cost of the control sets in that cost group (**ETC-Ada**). We also investigated simple extensions of TS-PSQ, UCB-PSQ, and expected improvement (adapted for BOPSQ) for the BOCVS problem by dividing the acquisition score of a control set by its cost in a manner similar to that in [31, Sec. 3.2]. We observed that these naive methods generally do not work well; we defer the results and discussion of these methods to Appendix B. Refer to Appendix C for full descriptions of all experimental settings and algorithm hyperparameters. The code for the experiments may be found at `https://github.com/sebtsh/bocvs`.

Fig. 2 shows the mean and standard error (over 10 RNG seeds) of the simple regret $\min_{1 \leq t \leq \mathcal{T}(C)} \mathbb{E}\big[f([\mathbf{x}^{i^*}, \mathbf{X}^{-i^*}])\big] - \mathbb{E}\big[f([\mathbf{x}^{i_t}, \mathbf{X}^{-i_t}])\big]$ (lower is better) incurred against cost spent (budget) $C$ by each algorithm with varying objective functions, cost sets, and variances of distributions where $\mathcal{T}(C)$ denotes the maximum iteration reached after spending $C$. The simple regret encodes the value of the best solution an algorithm has chosen within a certain budget and is a measure of cost efficiency. We report the salient observations below:

**(1) UCB-CVS variants outperform TS-PSQ and UCB-PSQ under cheap/moderate costs when the full query control set is available.** With the GP sample, Hartmann, and plant growth objective functions, the full query control set is available. TS-PSQ and UCB-PSQ only choose the full query control set in every iteration and are very cost inefficient under cheap and moderate costs, while UCB-CVS variants are able to use the cheaper control sets for exploration, followed by using the full query control set for exploitation, and find much better solutions with the same budget. As expected, their performance advantage reduces as the costs increase and $c_m$ gets closer to $c_i$ for all $i \neq m$.

**(2) Cost-adaptive UCB-CVS (ETC-Ada) can maintain competitive performance under expensive costs.** The non-cost-adaptive variants, ETC-50 and ETC-100, perform worse than TS-PSQ and UCB-PSQ under expensive costs. In contrast, it can be observed that ETC-Ada generally performs well under all costs by tuning the number of plays of suboptimal cost groups according to their costs. We recommend practitioners to use adaptive algorithms to achieve good performance under any cost set. In particular, the results suggest that an $\mathcal{O}(c_i^{-1})$ threshold is likely to work well across different sets of costs and is a robust choice for practitioners that keeps the number of hyperparameters to a minimum.

**(3) TS-PSQ and UCB-PSQ perform relatively well when the control sets are not subsets of each other.** With the airfoil self-noise objective function, TS-PSQ and UCB-PSQ perform better as the

control sets with this objective function are not subsets of each other and thus, they can also use the cheaper control sets during learning, while the UCB-CVS variants suffer worse performance here due to artificially selecting suboptimal control sets and queries with the $\epsilon$-relaxations. This worse performance is encoded in Theorems 4.1 and 4.4 as the sum of $\epsilon_t$ terms.

**(4) Increasing the variance of the probability distributions has competing effects on the simple regret.** Of the 42 experimental settings (combinations of objective function, cost set, and algorithm) in which the variance makes a difference (excluding TS-PSQ and UCB-PSQ for all objective functions except airfoil), the settings with variance 0.02, 0.04, and 0.08 achieved the lowest mean simple regret by the end 11, 6, and 25 times, respectively. This generally supports Theorem 4.4's prediction that higher variances decrease the upper bound on regret. However, due to the looseness of the bound, this effect is not guaranteed and there are still cases where lower variances lead to a lower regret, as suggested by the argument about feasible sets when discussing Theorem 4.1; note that the same MIGs of the feasible sets for control sets 1 to $m-1$ appear in Theorem 4.4. We observe competing effects and conclude that the effect of increasing variance is problem- and algorithm-dependent. While higher variances may lead to more exploration, they may also result in too much smoothing of function values which may hinder the learner's ability to focus on high-value query regions.

## 6 Conclusion

This paper introduces the BOCVS problem and describes the UCB-CVS algorithm that is provably no-regret in solving this problem. We show that our algorithm performs well across several different experimental settings and achieves the desired goal of finding significantly better solutions within the same budget. This work opens up avenues of future research: In particular, an entropy search-based algorithm [14, 23, 41] that chooses control sets and queries based on expected information gain per unit cost is a non-trivial and promising direction for alternative methods of solving BOCVS.

## Acknowledgements and Disclosure of Funding

This research/project is supported by the Agency for Science, Technology and Research, Singapore (A*STAR), under its RIE2020 Advanced Manufacturing and Engineering (AME) Programmatic Funds (Award A20H6b0151) and its RIE2020 Advanced Manufacturing and Engineering (AME) Industry Alignment Fund – Pre Positioning (IAF-PP) (Award A19E4a0101). Sebastian Shenghong Tay is supported by A*STAR.

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

# A  Proofs

## A.1  Proof of Theorem 4.1

*Theorem* 4.1. *With probability at least* $1 - \delta$, *UCB-CVS (Alg. 1) incurs a cost-varying cumulative regret bounded by*

$$R_T \leq \mathcal{O}\left( \left( B + \sqrt{\gamma_T(\mathcal{X}) + \log \frac{m+1}{\delta}} \right) \left( \sum_{i=1}^{m} c_i \left( \sqrt{T_i \gamma_{T_i}(\widetilde{\mathcal{X}}_i)} + \log \frac{m+1}{\delta} \right) \right) \right) + c_m \sum_{t=1}^{T} \epsilon_t.$$

*by setting* $\beta_t = B + \sigma \sqrt{2 \left( \gamma_{t-1}(\mathcal{X}) + 1 + \log((m+1)/\delta) \right)}$.

*Proof.*

$$R_T := \sum_{t=1}^{T} c_{i_t} \left( \mathbb{E}\left[ f([\mathbf{x}^{i^*}, \mathbf{X}^{-i^*}]) \right] - \mathbb{E}\left[ f([\mathbf{x}^{i_t}, \mathbf{X}^{-i_t}]) \right] \right)$$

$$\leq \sum_{t=1}^{T} c_{i_t} \left( \mathbb{E}\left[ u_{t-1}([\mathbf{x}^{i^*}, \mathbf{X}^{-i^*}]) \right] - \mathbb{E}\left[ f([\mathbf{x}^{i_t}, \mathbf{X}^{-i_t}]) \right] \right)$$

$$\stackrel{(i)}{\leq} \sum_{t=1}^{T} c_{i_t} \left( \mathbb{E}\left[ u_{t-1}([\mathbf{x}^{i_t}, \mathbf{X}^{-i_t}]) \right] - \mathbb{E}\left[ f([\mathbf{x}^{i_t}, \mathbf{X}^{-i_t}]) \right] + \epsilon_t \right)$$

$$\leq \sum_{t=1}^{T} c_{i_t} \left( \mathbb{E}\left[ u_{t-1}([\mathbf{x}^{i_t}, \mathbf{X}^{-i_t}]) \right] - \mathbb{E}\left[ f([\mathbf{x}^{i_t}, \mathbf{X}^{-i_t}]) \right] \right) + c_m \sum_{t=1}^{T} \epsilon_t \tag{3}$$

$$= \sum_{t=1}^{T} c_{i_t} \left( \mathbb{E}\left[ u_{t-1}([\mathbf{x}^{i_t}, \mathbf{X}^{-i_t}]) - f([\mathbf{x}^{i_t}, \mathbf{X}^{-i_t}]) \right] \right) + c_m \sum_{t=1}^{T} \epsilon_t$$

$$= \left( \sum_{i=1}^{m} c_i \sum_{t \in \widetilde{T}_i} \mathbb{E}\left[ u_{t-1}([\mathbf{x}^{i_t}, \mathbf{X}^{-i_t}]) - f([\mathbf{x}^{i_t}, \mathbf{X}^{-i_t}]) \right] \right) + c_m \sum_{t=1}^{T} \epsilon_t$$

$$\stackrel{(ii)}{\leq} \left( \sum_{i=1}^{m} c_i (2\beta_T) \sum_{t \in \widetilde{T}_i} \mathbb{E}\left[ \sigma_{t-1}([\mathbf{x}^{i_t}, \mathbf{X}^{-i_t}]) \right] \right) + c_m \sum_{t=1}^{T} \epsilon_t$$

$$\stackrel{(iii)}{\leq} \left( \sum_{i=1}^{m} c_i (2\beta_T) \left( 2 \sum_{t \in \widetilde{T}_i} \sigma_{t-1}(\mathbf{x}_t) + 4 \log \frac{m+1}{\delta} + 8 \log(4) + 1 \right) \right) + c_m \sum_{t=1}^{T} \epsilon_t \tag{4}$$

$$\stackrel{(iv)}{\leq} \left( \sum_{i=1}^{m} c_i (2\beta_T) \left( 2 \sqrt{4(T_i + 2) \gamma_{T_i}(\widetilde{\mathcal{X}}_i)} + 4 \log \frac{m+1}{\delta} + 8 \log(4) + 1 \right) \right) + c_m \sum_{t=1}^{T} \epsilon_t$$

$$= \mathcal{O}\left( \sum_{i=1}^{m} c_i \beta_T \left( \sqrt{T_i \gamma_{T_i}(\widetilde{\mathcal{X}}_i)} + \log \frac{m+1}{\delta} \right) \right) + c_m \sum_{t=1}^{T} \epsilon_t \tag{5}$$

$$= \mathcal{O}\left( \left( B + \sqrt{\gamma_T(\mathcal{X}) + \log \frac{m+1}{\delta}} \right) \left( \sum_{i=1}^{m} c_i \left( \sqrt{T_i \gamma_{T_i}(\widetilde{\mathcal{X}}_i)} + \log \frac{m+1}{\delta} \right) \right) \right) + c_m \sum_{t=1}^{T} \epsilon_t$$

where $\widetilde{T}_i$ is the ordered sequence of iterations at which control set $i$ is chosen, $(i)$ follows from the algorithm's choice of $\mathbf{x}^{i_t}$, $(ii)$ follows from Lemma A.4 with probability $\delta/(m+1)$, $(iii)$ follows from Lemma A.1 with probability $\delta/(m+1)$ applied once for each control set $i$, and $(iv)$ follows from Lemma A.3 and the definition of $\widetilde{\mathcal{X}}_i$ as the feasible set for control set $i$. A union bound over the $m+1$ events comprising the $m$ applications of Lemma A.1 and single application of Lemma A.4 yields the desired $1 - \delta$ probability bound. $\qquad\square$

## A.2 Proof of Lemma A.1

**Lemma A.1.** *Let $k(\mathbf{x}, \mathbf{x}) = 1$ and let $\widetilde{T}_i$ be the ordered sequence of iterations at which control set $i$ is chosen by UCB-CVS. For any $i \in [m]$, with probability at least $1 - \delta$,*

$$\sum_{t \in \widetilde{T}_i} \mathbb{E}\big[\sigma_{t-1}([\mathbf{x}^{i_t}, \mathbf{X}^{-i_t}])\big] \leq 2 \sum_{t \in \widetilde{T}_i} \sigma_{t-1}(\mathbf{x}_t) + 4 \log \frac{1}{\delta} + 8 \log 4 + 1.$$

*Proof.* For this proof, define a probability space $(\Omega, \mathcal{F}, \mathcal{P})$ and a filtration $\mathbb{F} = \{\mathcal{F}_t\}_{t=1}^{\infty}$, where $\mathcal{F}_t := \sigma(i_1, \mathbf{x}^{i_1}, \mathbf{x}^{-i_1}, y_1, i_2, \mathbf{x}^{i_2}, \mathbf{x}^{-i_2}, y_2, ..., i_t, \mathbf{x}^{i_t}, \mathbf{x}^{-i_t}, y_t)$, the sigma-algebra generated by all the random variables in the BO procedure known by the end of iteration $t$.

In advance of proving this result, it should be clarified that, in the main paper and all proofs excluding that of this lemma, $\mathbb{E}\big[\sigma_{t-1}([\mathbf{x}^{i_t}, \mathbf{X}^{-i_t}])\big]$ denotes the quantity obtained by treating $\sigma_{t-1}$ and $\mathbf{x}^{i_t}$ as deterministic and treating $\mathbf{X}^{-i_t}$ as a random vector distributed according to the probability distribution $\mathbb{P}^{-i_t}$. This is for ease of exposition. In the following proof, however, when using the formalism of random processes, what was previously referred to as $\mathbb{E}\big[\sigma_{t-1}([\mathbf{x}^{i_t}, \mathbf{X}^{-i_t}])\big]$ is actually $\mathbb{E}[\sigma_{t-1}(\mathbf{x}_t) \,|\, \mathcal{F}_{t-1}]$. The meaning is equivalent since $\sigma_{t-1}$, $i_t$, and $\mathbf{x}^{i_t}$ are $\mathbb{F}$-predictable, and the only uncertainty about $\sigma_{t-1}(\mathbf{x}_t)$ arises from the lack of knowledge about $\mathbf{x}^{-i_t}$.

Now we begin the proof proper. Define $m$ stochastic processes $\{X_t^{(1)}\}_{t=1}^{\infty}$, $\{X_t^{(2)}\}_{t=1}^{\infty}$, ..., $\{X_t^{(m)}\}_{t=1}^{\infty}$, where, using $\mathbb{1}[A]$ to denote the indicator function that is equal to 1 when the event $A$ is true and 0 otherwise,

$$X_t^{(i)} := \sigma_{t-1}(\mathbf{x}_t) \cdot \mathbb{1}[i_t = i] \ .$$

Since each $X_t^{(i)}$ is $\mathcal{F}_t$-measurable, each stochastic process is adapted to $\mathbb{F}$. Next, define

$$\begin{aligned}
m_t^{(i)} &:= \mathbb{E}\Big[X_t^{(i)} \,|\, \mathcal{F}_{t-1}\Big] \\
&= \mathbb{E}[\sigma_{t-1}(\mathbf{x}_t) \cdot \mathbb{1}[i_t = i] \,|\, \mathcal{F}_{t-1}] \\
&= \mathbb{1}[i_t = i] \, \mathbb{E}[\sigma_{t-1}(\mathbf{x}_t) \,|\, \mathcal{F}_{t-1}]
\end{aligned}$$

where the last equality uses the pull-through property since $i_t$ is $\mathcal{F}_{t-1}$ measurable. Using Lemma A.5 with $b_t = 1$ since $k(\mathbf{x}, \mathbf{x}) = 1$, with probability at least $1 - \delta$, for any $T \geq 1$,

$$\sum_{t=1}^{T} m_t^{(i)} \leq 2 \sum_{t=1}^{T} X_t^{(i)} + 4 \log \frac{1}{\delta} + 8 \log 4 + 1$$

$$\sum_{t=1}^{T} \mathbb{1}[i_t = i] \, \mathbb{E}[\sigma_{t-1}(\mathbf{x}_t) \,|\, \mathcal{F}_{t-1}] \leq 2 \sum_{t=1}^{T} \sigma_{t-1}(\mathbf{x}_t) \cdot \mathbb{1}[i_t = i] + 4 \log \frac{1}{\delta} + 8 \log 4 + 1$$

$$\sum_{t \in \widetilde{T}_i} \mathbb{E}[\sigma_{t-1}(\mathbf{x}_t) \,|\, \mathcal{F}_{t-1}] \leq 2 \sum_{t \in \widetilde{T}_i} \sigma_{t-1}(\mathbf{x}_t) + 4 \log \frac{1}{\delta} + 8 \log 4 + 1$$

which completes the proof. $\qquad\square$

## A.3 Proof of Proposition 4.2

*Proposition* 4.2. *If there exists a $\tilde{\epsilon} > 0$ s.t. for all $i \neq i^*$,*

$$\epsilon_t \leq \mathbb{E}\Big[f([\mathbf{x}^{i^*}, \mathbf{X}^{-i^*}])\Big] - \max_{\mathbf{x}^i \in \mathcal{X}^i} \mathbb{E}\Big[f([\mathbf{x}^i, \mathbf{X}^{-i}])\Big] - \tilde{\epsilon}$$

*eventually (i.e., the inequality holds for all $t \geq q$ for some $q \geq 1$), and $\gamma_T(\mathcal{X}) < \mathcal{O}(\sqrt{T})$, then, with probability at least $1 - \delta$, $\lim_{T \to \infty} T_i/T = 0$ for all $i \neq i^*$ and UCB-CVS incurs a cost-varying cumulative regret that is sublinear in $T$ by setting $\beta_t = B + \sigma\sqrt{2\left(\gamma_{t-1}(\mathcal{X}) + 1 + \log((m+1)/\delta)\right)}$.*

*Proof.* Define $\mathbf{x}_t^i := \operatorname{argmax}_{\mathbf{x}^i \in \mathcal{X}^i} \mathbb{E}\big[u_{t-1}([\mathbf{x}^i, \mathbf{X}^{-i}])\big]$, and $j_t := \operatorname{argmax}_{i \in [m]} \max_{\mathbf{x}^i \in \mathcal{X}^i} \mathbb{E}\big[u_{t-1}([\mathbf{x}^i, \mathbf{X}^{-i}])\big]$. Using $\mathbb{1}[A]$ to denote the indicator function

that is equal to 1 when the event $A$ is true and 0 otherwise,

$$T_i \overset{(i)}{\leq} \sum_{t=1}^{T} \mathbb{1}\Big[\mathbb{E}\big[u_{t-1}([\mathbf{x}_t^i, \mathbf{X}^{-i}])\big] + \epsilon_t \geq \mathbb{E}\big[u_{t-1}([\mathbf{x}_t^{j_t}, \mathbf{X}^{-j_t}])\big]\Big]$$

$$\overset{(ii)}{\leq} \sum_{t=1}^{T} \mathbb{1}\Big[\mathbb{E}\big[u_{t-1}([\mathbf{x}_t^i, \mathbf{X}^{-i}])\big] + \epsilon_t \geq \mathbb{E}\big[u_{t-1}([\mathbf{x}^{i^*}, \mathbf{X}^{-i^*}])\big]\Big]$$

$$\leq \sum_{t=1}^{T} \mathbb{1}\Big[\mathbb{E}\big[u_{t-1}([\mathbf{x}_t^i, \mathbf{X}^{-i}])\big] + \epsilon_t \geq \mathbb{E}\big[f([\mathbf{x}^{i^*}, \mathbf{X}^{-i^*}])\big]\Big]$$

$$= \sum_{t=1}^{T} \mathbb{1}\Big[\mathbb{E}\big[u_{t-1}([\mathbf{x}_t^i, \mathbf{X}^{-i}])\big] \geq \mathbb{E}\big[f([\mathbf{x}^{i^*}, \mathbf{X}^{-i^*}])\big] - \epsilon_t\Big]$$

$$\leq q - 1 + \sum_{t=q}^{T} \mathbb{1}\Big[\mathbb{E}\big[u_{t-1}([\mathbf{x}_t^i, \mathbf{X}^{-i}])\big] \geq \mathbb{E}\big[f([\mathbf{x}^{i^*}, \mathbf{X}^{-i^*}])\big] - \epsilon_t\Big]$$

$$\leq q - 1 + \sum_{t=q}^{T} \mathbb{1}\Big[\mathbb{E}\big[u_{t-1}([\mathbf{x}_t^i, \mathbf{X}^{-i}])\big] \geq \max_{\mathbf{x}^i \in \mathcal{X}^i} \mathbb{E}\big[f([\mathbf{x}^i, \mathbf{x}^{-i}])\big] + \tilde{\epsilon}\Big]$$

$$\leq q - 1 + \sum_{t=q}^{T} \mathbb{1}\Big[\mathbb{E}\big[u_{t-1}([\mathbf{x}_t^i, \mathbf{X}^{-i}])\big] \geq \mathbb{E}\big[f([\mathbf{x}_t^i, \mathbf{X}^{-i}])\big] + \tilde{\epsilon}\Big]$$

$$= q - 1 + \sum_{t=q}^{T} \mathbb{1}\Big[\mathbb{E}\big[u_{t-1}([\mathbf{x}_t^i, \mathbf{X}^{-i}])\big] - \mathbb{E}\big[f([\mathbf{x}_t^i, \mathbf{X}^{-i}])\big] \geq \tilde{\epsilon}\Big]$$

$$\leq q - 1 + \frac{1}{\tilde{\epsilon}} \sum_{t=q}^{T} \mathbb{E}\big[u_{t-1}([\mathbf{x}_t^i, \mathbf{X}^{-i}])\big] - \mathbb{E}\big[f([\mathbf{x}_t^i, \mathbf{X}^{-i}])\big]$$

$$\overset{(iii)}{\leq} q - 1 + \mathcal{O}^*\left(\frac{1}{\tilde{\epsilon}}\sqrt{T-q+1}\left(B\sqrt{\gamma_{T-q+1}(\mathcal{X})} + \gamma_{T-q+1}(\mathcal{X})\right)\right)$$

where $(i)$ follows since control set $i$ is only played when the condition on the RHS is true, $(ii)$ follows from the definitions of $j_t$ and $\mathbf{x}_t^{j_t}$, and $(iii)$ follows from the steps from (3) to (5) in the proof of Theorem 4.1, and $\mathcal{O}^*$ denotes suppressing logarithmic factors. Now dividing both sides by $T$ and taking the limit as $T$ goes to infinity,

$$\lim_{T\to\infty} \frac{T_i}{T} \leq \lim_{T\to\infty} \frac{1}{T}\left(q - 1 + \mathcal{O}^*\left(\frac{1}{\tilde{\epsilon}}\sqrt{T-q+1}\left(B\sqrt{\gamma_{T-q+1}(\mathcal{X})} + \gamma_{T-q+1}(\mathcal{X})\right)\right)\right)$$
$$= 0$$

which follows from $\gamma_T(\mathcal{X}) < \mathcal{O}(\sqrt{T})$ and completes the proof that, if the conditions in the proposition are fulfilled, suboptimal control sets will only be played a number of times that is sublinear in $T$. The proof that $R_T$ will then also be sublinear in $T$ is straightforward. Assuming without loss of generality that $i^* = m$,

$$R_T := \sum_{t=1}^{T} c_{i_t}\left(\mathbb{E}\big[f([\mathbf{x}^{i^*}, \mathbf{X}^{-i^*}])\big] - \mathbb{E}\big[f([\mathbf{x}^{i_t}, \mathbf{X}^{-i_t}])\big]\right)$$

$$= \sum_{i=1}^{m} \sum_{t\in\widetilde{T}_i} c_i\left(\mathbb{E}\big[f([\mathbf{x}^{i^*}, \mathbf{X}^{-i^*}])\big] - \mathbb{E}\big[f([\mathbf{x}^{i_t}, \mathbf{X}^{-i_t}])\big]\right)$$

$$\leq \sum_{i=1}^{m-1} T_i c_i\left(\mathbb{E}\big[f([\mathbf{x}^{i^*}, \mathbf{X}^{-i^*}])\big] - \min_{\mathbf{x}^i \in \mathcal{X}^i} \mathbb{E}\big[f([\mathbf{x}^i, \mathbf{X}^{-i}])\big]\right)$$

$$+ c_m \sum_{t\in\widetilde{T}_m}\left(\mathbb{E}\big[f([\mathbf{x}^{i^*}, \mathbf{X}^{-i^*}])\big] - \mathbb{E}\big[f([\mathbf{x}^{i_t}, \mathbf{X}^{-i_t}])\big]\right)$$

$$= \sum_{i=1}^{m-1} T_i C_i + c_m \sum_{t \in \widetilde{T}_m} \left( \mathbb{E}\left[ f([\mathbf{x}^{i^*}, \mathbf{X}^{-i^*}]) \right] - \mathbb{E}\left[ f([\mathbf{x}^{i_t}, \mathbf{X}^{-i_t}]) \right] \right)$$

$$\overset{(i)}{\leq} \sum_{i=1}^{m-1} T_i C_i + \mathcal{O}\left( c_m \left( B + \sqrt{\gamma_T(\mathcal{X}) + \log \frac{m+1}{\delta}} \right) \left( \sqrt{T_m \gamma_{T_m}(\widetilde{\mathcal{X}}_m)} + \log \frac{m+1}{\delta} \right) \right)$$

where $\widetilde{T}_i$ is the ordered sequence of iterations at which control set $i$ is chosen and $C_i :=$ $c_i \left( \mathbb{E}\left[ f([\mathbf{x}^{i^*}, \mathbf{X}^{-i^*}]) \right] - \min_{\mathbf{x}^i \in \mathcal{X}^i} \mathbb{E}\left[ f([\mathbf{x}^i, \mathbf{X}^{-i}]) \right] \right)$, and $(i)$ follows from the steps in the proof of Theorem 4.1 but only for control set $m$ and without accounting for the $\epsilon$-schedule. Since each $T_i$ is sublinear in $T$, dividing both sides by $T$, using the fact that $\gamma_T(\mathcal{X}) < \mathcal{O}(\sqrt{T})$, and taking the limit as $T \to \infty$ yields the desired result and completes the proof. $\qquad\square$

## A.4  Proof of Lemma 4.3

*Lemma* 4.3. *Assume that, for each control set $i$, the random variable $Y_i := \left\| [\mathbf{0}, \mathbf{X}_1^{-i}] - [\mathbf{0}, \mathbf{X}_2^{-i}] \right\|^2$ has a median $M_i$ such that $\mathbb{E}[Y_i | Y_i > M_i] \leq h_i M_i$ for some $h_i > 0$, where $\mathbf{X}_1^{-i}, \mathbf{X}_2^{-i} \sim \mathbb{P}^{-i}$. With probability at least $1 - \delta$, there will be at least $N_i$ non-overlapping pairs of queries $\mathbf{x}$ and $\mathbf{x}'$ chosen by UCB-CVS (Alg. 1) with control set $i$ such that $\|\mathbf{x} - \mathbf{x}'\|^2 \geq M_i$, where*

$$N_i = \left\lfloor \frac{1}{4}(T_i - 1) - \sqrt{\frac{1}{4}T_i \log \frac{1}{\delta}} \right\rfloor,$$

$$M_i \geq \frac{4}{h_i + 1} \sum_{\ell \in \overline{\mathcal{I}}_i} \mathbb{V}[X_\ell].$$

*Proof.* Consider two queries $\mathbf{x} = [\mathbf{x}^i, \mathbf{x}^{-i}]$ and $\mathbf{x}' = [\mathbf{x}'^i, \mathbf{x}'^{-i}]$ chosen with control set $i$. The learner only selects $\mathbf{x}^i$ and $\mathbf{x}'^i$ while $\mathbf{x}^{-i}$ and $\mathbf{x}'^{-i}$ are sampled from the environment. Before they are sampled, the queries may be considered themselves random vectors composed of one deterministic partial query and one random partial query. Denote these random vectors as $\mathbf{X} = [\mathbf{x}^i, \mathbf{X}^{-i}]$ and $\mathbf{X}' = [\mathbf{x}'^i, \mathbf{X}'^{-i}]$. $\|\mathbf{X} - \mathbf{X}'\|^2$ is therefore a random variable as well. Observe that

$$\|\mathbf{X} - \mathbf{X}'\|^2 = \sum_{j \in \mathcal{I}_i} (x_j - x_j')^2 + \sum_{\ell \in \overline{\mathcal{I}}_i} (X_\ell - X_\ell')^2$$

$$\geq \sum_{\ell \in \overline{\mathcal{I}}_i} (X_\ell - X_\ell')^2$$

$$= \left\| [\mathbf{0}, \mathbf{X}_1^{-i}] - [\mathbf{0}, \mathbf{X}_2^{-i}] \right\|^2$$

$$= Y_i'.$$

where $Y_i'$ is a random variable that is i.i.d. with $Y_i$. Therefore, any $\|\mathbf{X} - \mathbf{X}'\|^2$ can be treated as a random variable equal to some $Y_i'$ that is i.i.d. with $Y_i$ plus some non-negative term. The rest of this proof will use lower bounds on random variables i.i.d. with $Y_i$, which will in turn imply lower bounds on $\|\mathbf{X} - \mathbf{X}'\|^2$.

$$\mathbb{E}[Y_i] = \mathbb{E}\left[ \sum_{\ell \in \mathcal{I}_i} (X_\ell - X_\ell')^2 \right]$$

$$= \sum_{\ell \in \overline{\mathcal{I}}_i} \mathbb{E}\left[ (X_\ell - \overline{X}_\ell - (X_\ell' - \overline{X}_\ell))^2 \right]$$

$$= \sum_{\ell \in \overline{\mathcal{I}}_i} \mathbb{E}\left[ ((X_\ell - \overline{X}_\ell) - (X_\ell' - \overline{X}_\ell'))^2 \right]$$

$$= \sum_{\ell \in \overline{\mathcal{I}}_i} \mathbb{E}\left[ (X_\ell - \overline{X}_\ell)^2 - 2(X_\ell - \overline{X}_\ell)(X_\ell' - \overline{X}_\ell') + (X_\ell' - \overline{X}_\ell')^2 \right]$$

$$= \sum_{\ell \in \overline{\mathcal{I}}_i} \mathbb{E}\big[(X_\ell - \overline{X}_\ell)^2\big] - \mathbb{E}\Big[(X_\ell - \overline{X}_\ell)(X'_\ell - \overline{X}'_\ell)\Big] + \mathbb{E}\Big[(X'_\ell - \overline{X}'_\ell)^2\Big]$$

$$= \sum_{\ell \in \overline{\mathcal{I}}_i} \mathbb{E}\big[(X_\ell - \overline{X}_\ell)^2\big] + \mathbb{E}\Big[(X'_\ell - \overline{X}'_\ell)^2\Big]$$

$$= \sum_{\ell \in \overline{\mathcal{I}}_i} 2\mathbb{V}[X_\ell]. \tag{6}$$

We will now construct a lower bound for a median of $Y_i$ denoted $M_i$.

$$\mathbb{E}[Y_i] = \mathbb{E}[Y_i|Y_i < M_i] \cdot P(Y_i < M_i) + \mathbb{E}[Y_i|Y_i = M_i] \cdot P(Y_i = M_i) + \mathbb{E}[Y_i|Y_i > M_i] \cdot P(Y_i > M_i)$$

$$\leq M_i \cdot P(Y_i \leq M_i) + \mathbb{E}[Y_i|Y_i > M_i] \cdot P(Y_i > M_i)$$

$$\overset{(i)}{\leq} M_i \cdot P(Y_i \leq M_i) + h_i M_i \cdot P(Y_i > M_i)$$

$$\overset{(ii)}{\leq} \frac{1}{2}M_i + \frac{1}{2}(h_i \cdot M_i)$$

$$= \frac{h_i + 1}{2}M_i$$

where $(i)$ follows from our assumption on the median $M_i$ and $(ii)$ follows from the definition of a median: $P(Y_i \leq M_i) \geq 1/2$. Substituting in (6) completes our construction of the lower bound for $M_i$:

$$M_i \geq \frac{2}{h_i + 1}\mathbb{E}[Y_i]$$

$$\geq \frac{4}{h_i + 1} \sum_{\ell \in \overline{\mathcal{I}}_i} \mathbb{V}[X_\ell].$$

Now consider the $\lfloor T_i/2 \rfloor$ non-overlapping pairs of queries chosen with control set $i$ [3]. Associate each pair with a random variable $Y_{ij}$ such that we have $\lfloor T_i/2 \rfloor$ i.i.d. random variables $Y_{i1}, Y_{i2}, ..., Y_{i\lfloor T_i/2 \rfloor}$. From the definition of a median, $P(Y_i \geq M_i) \geq 1/2$. Without loss of generality, assume the worst-case such that $P(Y_i \geq M_i) = 1/2$. We can now construct $\lfloor T_i/2 \rfloor$ i.i.d. Bernoulli random variables $Z_1, Z_2, ..., Z_n, n = \lfloor T_i/2 \rfloor$, with $p = 1/2$ where a success $(Z_j = 1)$ corresponds to $Y_{ij} \geq M_i$ and a failure $(Z_j = 0)$ corresponds to $Y_{ij} < M_i$. Further define the random variable $Z := \sum_{j=1}^n Z_j$.

Applying Hoeffding's inequality,

$$P\left(\frac{1}{n}\sum_{j=1}^n (Z_j - p) \leq -t\right) \leq \exp\left(-2nt^2\right)$$

$$P\left(\frac{1}{n}Z - p \leq -t\right) \leq \exp\left(-2nt^2\right)$$

$$P\left(Z \leq n(p - t)\right) \leq \exp\left(-2nt^2\right).$$

Choosing $t = p - \alpha/n$ for some constant $\alpha$,

$$P\left(Z \leq \alpha\right) \leq \exp\left(-2n\left(p - \frac{\alpha}{n}\right)^2\right).$$

For $P\left(Z \leq \alpha\right) \leq \delta$,

$$\exp\left(-2n\left(p - \frac{\alpha}{n}\right)^2\right) = \delta$$

---

[3]While we technically have $\binom{T_i}{2}$ (overlapping) pairs, the squared distances between each such pair will be identically distributed but not independent. For example, if $T_i \geq 3$ and we knew that $\binom{T_i}{2} - 1$ of the squared distances were equal to 0 (i.e., all the queries are exactly the same), the last squared distance must also be equal to 0.

$$\alpha = np - \sqrt{\frac{n}{2}\log\frac{1}{\delta}}$$

$$\alpha = \frac{1}{2}\left\lfloor\frac{T_i}{2}\right\rfloor - \sqrt{\frac{1}{2}\left\lfloor\frac{T_i}{2}\right\rfloor\log\frac{1}{\delta}}$$

$$\alpha \geq \frac{1}{4}(T_i - 1) - \sqrt{\frac{1}{4}T_i\log\frac{1}{\delta}}$$

$$\alpha \geq \left\lfloor\frac{1}{4}(T_i - 1) - \sqrt{\frac{1}{4}T_i\log\frac{1}{\delta}}\right\rfloor.$$

Therefore, with probability more than $1-\delta$, $Z > N_i := \left\lfloor\frac{1}{4}(T_i-1) - \sqrt{\frac{1}{4}T_i\log\frac{1}{\delta}}\right\rfloor$, i.e., the number of non-overlapping pairs with squared distance greater than $M_i$ is at least $N_i$, which completes the proof. $\qquad\square$

### A.5 Proof of Theorem 4.4

*Theorem* 4.4. *If the following assumptions hold:*

1. *The assumption of Lemma 4.3 holds;*

2. *The kernel $k(\mathbf{x}, \mathbf{x}')$ is an isotropic kernel (which only depends on distance and can be written as $k(\|\mathbf{x}-\mathbf{x}'\|)$);*

3. *There exists an iteration $r$ such that for all $t \leq r$, $i_t \leq m-1$ and for all $t > r$, $i_t = m$;*

*then with probability at least $1-\delta$, UCB-CVS (Alg. 1) incurs a cost-varying cumulative regret bounded by*

$$R_T \leq c_m \sum_{t=1}^{T}\epsilon_t + \mathcal{O}\left(\left(B + \sqrt{\gamma_T(\mathcal{X}) + \log\frac{2m}{\delta}}\right)\left(c_m\left(\sqrt{T\gamma_T(\mathcal{X})} - \mathcal{L} + \log\frac{2m}{\delta}\right) + \sum_{i=1}^{m-1}c_i\left(\sqrt{T_i\gamma_{T_i}(\widetilde{\mathcal{X}}_i)} + \log\frac{2m}{\delta}\right)\right)\right)$$

$$\mathcal{L} := \lambda\left(\sum_{i=1}^{m-1}N_i\log\left(V_i - 2k\left(\sqrt{M_i}\right) - k\left(\sqrt{M_i}\right)^2\right) + W\right)$$

*by setting $\beta_t = B + \sigma\sqrt{2\left(\gamma_{t-1}(\mathcal{X}) + 1 + \log((2m)/\delta)\right)}$, where $N_i$ and $M_i$ are defined as in Lemma 4.3, and $V_i$ and $W$ are residual terms defined in (10).*

*Proof.* We first construct a lower bound on the sum of posterior standard deviations of the queries up to iteration $r$, i.e., the queries that were chosen with any control set except the last.

$$\sum_{t=1}^{r}\sigma_{t-1}(\mathbf{x}_t) \overset{(i)}{\geq} \sum_{t=1}^{r}\sigma_{t-1}^2(\mathbf{x}_t)$$

$$= \lambda\sum_{t=1}^{r}\lambda^{-1}\sigma_{t-1}^2(\mathbf{x}_t)$$

$$\overset{(ii)}{\geq} \lambda\sum_{t=1}^{r}\log(1+\lambda^{-1}\sigma_{t-1}^2(\mathbf{x}_t))$$

$$\overset{(iii)}{=} \lambda\log\left|\mathbf{I} + \lambda^{-1}\mathbf{K}_r\right|$$

$$= \lambda\log\left(\lambda^{-r}|\lambda\mathbf{I}+\mathbf{K}_r|\right)$$

$$= \lambda\left(-r\log\lambda + \log|\lambda\mathbf{I}+\mathbf{K}_r|\right)$$

$$\overset{(iv)}{\geq} \lambda(\log|\lambda\mathbf{I}+\mathbf{K}_r| - 2) \tag{7}$$

where $(i)$ follows from the assumption that $k(x, x) = 1$ which implies $\sigma_{t-1}(\mathbf{x}) \leq 1$ for all $\mathbf{x} \in \mathcal{X}$ and all $t \geq 1$, $(ii)$ follows since $\log(1 + x) \leq x$ for all $x > -1$, $(iii)$ follows from Lemma A.2, and $(iv)$ follows from $\lambda = 1 + \frac{2}{T}$ (Lemma A.4), noting that $T \geq r$, and taking $\lim_{r \to \infty} -r \log \lambda$.

From Lemma 4.3 with probability $\delta/(2m)$, there will be at least $N_i$ pairs of queries chosen with control set $i$ with squared distance at least $M_i$, where

$$N_i = \left\lfloor \frac{1}{4}(T_i - 1) - \sqrt{\frac{1}{4}T_i \log \frac{2m}{\delta}} \right\rfloor$$

$$M_i = \frac{4}{h_i + 1} \sum_{\ell \in \overline{\mathcal{I}}_i} \mathbb{V}[X_\ell]$$

Gather these $2\sum_{i=1}^{m-1} N_i$ queries in an ordered sequence $\mathcal{S}$ and keep paired queries adjacent to each other. The sequence should be ordered such that, for any control sets $i$ and $j$, if $i < j$, then queries chosen with $i$ should appear in the sequence before queries chosen with $j$. Denote as $\widetilde{\mathcal{T}}$ the ordered sequence of iterations at which each of these queries were chosen by the learner where the order corresponds to the order in $\mathcal{S}$. Using row and column swaps on $\mathbf{K}_r$, construct a new Gram matrix $\mathbf{K}_s$ such that, for all $j, \ell \leq 2\sum_{i=1}^{m-1} N_i$,

$$[\mathbf{K}_s]_{j\ell} = [\mathbf{K}_r]_{\widetilde{\mathcal{T}}_j \widetilde{\mathcal{T}}_\ell}.$$

In other words, we have simply reordered the underlying queries that result in the Gram matrix $\mathbf{K}_r$ to produce a new Gram matrix $\mathbf{K}_s$ such that the first $2\sum_{i=1}^{m-1} N_i$ rows (and columns) correspond to the $2N_i$ queries, and paired queries (that have at least $M_i$ squared distance between them) are kept in adjacent rows (and columns). Note that

$$[\lambda \mathbf{I} + \mathbf{K}_r]_{\widetilde{\mathcal{T}}_j \widetilde{\mathcal{T}}_\ell} = [\lambda \mathbf{I} + \mathbf{K}_s]_{j\ell}$$

i.e., the same row and column swap operations on $\lambda \mathbf{I} + \mathbf{K}_r$ result in $\lambda \mathbf{I} + \mathbf{K}_s$. Note that swapping the positions of two queries corresponds to a row swap and a column swap in the Gram matrix. We can thus conclude that

$$|\lambda \mathbf{I} + \mathbf{K}_r| = |\lambda \mathbf{I} + \mathbf{K}_s| \tag{8}$$

since determinants are invariant under an even number of row or column swaps.

Write $|\lambda \mathbf{I} + \mathbf{K}_s|$ as

$$|\lambda \mathbf{I} + \mathbf{K}_s| = \begin{bmatrix} \mathbf{A}_1 & \mathbf{B}_1 \\ \mathbf{C}_1 & \mathbf{D}_1 \end{bmatrix}$$

where $\mathbf{A}_1$ is a $2 \times 2$ matrix. Since $\mathbf{A}_1$ is invertible,

$$|\lambda \mathbf{I} + \mathbf{K}_s| = |\mathbf{A}_1| \left| \mathbf{D}_1 - \mathbf{C}_1 \mathbf{A}_1^{-1} \mathbf{B}_1 \right|$$

where $\mathbf{D}_1 - \mathbf{C}_1 \mathbf{A}_1^{-1} \mathbf{B}_1$ is the Schur complement of $\mathbf{A}_1$. Observe that

$$\mathbf{D}_1 - \mathbf{C}_1 \mathbf{A}_1^{-1} \mathbf{B}_1 = \lambda \mathbf{I} + \mathbf{K}_{s-2} - \mathbf{k}_{2,s-2}^\top (\mathbf{K}_2 + \lambda \mathbf{I})^{-1} \mathbf{k}_{2,s-2}$$

$$= \lambda \mathbf{I} + \hat{\mathbf{K}}_{s-2}$$

where $\mathbf{K}_2$ and $\mathbf{K}_{s-2}$ are the prior covariance matrices of the first 2 queries and last $r - 2$ queries respectively, $\mathbf{k}_{2,s-2}$ is the prior covariance between the first 2 queries and the last $r - 2$ queries, and $\hat{\mathbf{K}}_{s-2}$ is the posterior covariance matrix of the last $r - 2$ queries conditioned on observations at the first 2 queries. We can repeat this decomposition:

$$\lambda \mathbf{I} + \hat{\mathbf{K}}_{s-2} = \begin{bmatrix} \mathbf{A}_2 & \mathbf{B}_2 \\ \mathbf{C}_2 & \mathbf{D}_2 \end{bmatrix}$$

$$\left| \lambda \mathbf{I} + \hat{\mathbf{K}}_{s-2} \right| = |\mathbf{A}_2| \left| \mathbf{D}_2 - \mathbf{C}_2 \mathbf{A}_2^{-1} \mathbf{B}_2 \right|$$

$$\mathbf{D}_2 - \mathbf{C}_2 \mathbf{A}_2^{-1} \mathbf{B}_2 = \lambda \mathbf{I} + \hat{\mathbf{K}}_{s-4}$$

where $\hat{\mathbf{K}}_{s-4}$ is the posterior covariance matrix of the last $r - 4$ queries conditioned on observations at the first 4 queries, by the quotient property of the Schur complement [8]. Define $N := \sum_{i=1}^{m-1} N_i$. Performing this decomposition $N$ times yields

$$|\lambda \mathbf{I} + \mathbf{K}_s| = \prod_{j=1}^{N} |\mathbf{A}_j| \left| \lambda \mathbf{I} + \hat{\mathbf{K}}_{s-2N} \right|$$

where each $\mathbf{A}_j$ is the $2 \times 2$ posterior covariance matrix of a pair of queries chosen with some control set $i$ that have least $M_i$ squared distance between them conditioned on observations at the first $2(j-1)$ queries in the sequence, plus $\lambda \mathbf{I}$. From (7) and (8),

$$\sum_{t=1}^{r} \sigma_{t-1}(\mathbf{x}_t) \geq \lambda \left( \sum_{j=1}^{N} \log |\mathbf{A}_j| + \log \left| \lambda \mathbf{I} + \hat{\mathbf{K}}_{s-2N} \right| - 2 \right). \tag{9}$$

Let $\hat{\mathbf{x}}_j$ and $\hat{\mathbf{x}}_j'$ refer to the pair of queries associated with $\mathbf{A}_j$, and $\tilde{k}_j$ to the posterior covariance function conditioned on observations at the first $2(j-1)$ queries in the sequence. Define $\mathbf{k}_j$ and $\mathbf{k}_j'$ as the $\mathbb{R}^{2(j-1)}$ vectors of the prior covariance between the first $2(j-1)$ queries in the sequence and $\hat{\mathbf{x}}_j$ and $\hat{\mathbf{x}}_j'$ respectively. Further define $\mathbf{M}_j := \mathbf{K}_{2(j-1)} + \lambda \mathbf{I}$. Use $\langle \mathbf{u}, \mathbf{v} \rangle_{\mathbf{M}}$ to denote $\mathbf{u}^\top \mathbf{M} \mathbf{v}$, and $\|\mathbf{u}\|_{\mathbf{M}}$ to denote $\sqrt{\langle \mathbf{u}, \mathbf{u} \rangle_{\mathbf{M}}}$. Each $|\mathbf{A}_j|$ can be lower bounded as

$$|\mathbf{A}_j| = (\tilde{k}_j(\hat{\mathbf{x}}_j, \hat{\mathbf{x}}_j) + \lambda)(\tilde{k}_j(\hat{\mathbf{x}}_j', \hat{\mathbf{x}}_j') + \lambda) - \tilde{k}_j(\hat{\mathbf{x}}_j, \hat{\mathbf{x}}_j')^2$$

$$= \tilde{k}_j(\hat{\mathbf{x}}_j, \hat{\mathbf{x}}_j)\tilde{k}_j(\hat{\mathbf{x}}_j', \hat{\mathbf{x}}_j') + \lambda\tilde{k}_j(\hat{\mathbf{x}}_j, \hat{\mathbf{x}}_j) + \lambda\tilde{k}_j(\hat{\mathbf{x}}_j', \hat{\mathbf{x}}_j') + \lambda^2 - \tilde{k}_j(\hat{\mathbf{x}}_j, \hat{\mathbf{x}}_j')^2$$

$$\stackrel{(i)}{=} \left(1 - \|\mathbf{k}_j\|^2_{\mathbf{M}_j^{-1}}\right)\left(1 - \|\mathbf{k}_j'\|^2_{\mathbf{M}_j^{-1}}\right) + \lambda\left(1 - \|\mathbf{k}_j\|^2_{\mathbf{M}_j^{-1}}\right) + \lambda\left(1 - \|\mathbf{k}_j'\|^2_{\mathbf{M}_j^{-1}}\right) + \lambda^2$$

$$\qquad - \left(k(\hat{\mathbf{x}}_j, \hat{\mathbf{x}}_j') - \langle \mathbf{k}_j, \mathbf{k}_j'\rangle_{\mathbf{M}_j^{-1}}\right)^2$$

$$= \left(1 - \|\mathbf{k}_j\|^2_{\mathbf{M}_j^{-1}}\right)\left(1 - \|\mathbf{k}_j'\|^2_{\mathbf{M}_j^{-1}}\right) + \lambda\left(1 - \|\mathbf{k}_j\|^2_{\mathbf{M}_j^{-1}}\right) + \lambda\left(1 - \|\mathbf{k}_j'\|^2_{\mathbf{M}_j^{-1}}\right) + \lambda^2$$

$$\qquad - k(\hat{\mathbf{x}}_j, \hat{\mathbf{x}}_j')^2 + 2k(\hat{\mathbf{x}}_j, \hat{\mathbf{x}}_j') \langle \mathbf{k}_j, \mathbf{k}_j'\rangle_{\mathbf{M}_j^{-1}} - \langle \mathbf{k}_j, \mathbf{k}_j'\rangle^2_{\mathbf{M}_j^{-1}}$$

$$\stackrel{(ii)}{\geq} \left(1 - \|\mathbf{k}_j\|^2_{\mathbf{M}_j^{-1}}\right)\left(1 - \|\mathbf{k}_j'\|^2_{\mathbf{M}_j^{-1}}\right) + \lambda\left(1 - \|\mathbf{k}_j\|^2_{\mathbf{M}_j^{-1}}\right) + \lambda\left(1 - \|\mathbf{k}_j'\|^2_{\mathbf{M}_j^{-1}}\right) + \lambda^2$$

$$\qquad - k(\hat{\mathbf{x}}_j, \hat{\mathbf{x}}_j')^2 - 2k(\hat{\mathbf{x}}_j, \hat{\mathbf{x}}_j') \|\mathbf{k}_j\|_{\mathbf{M}_j^{-1}} \|\mathbf{k}_j'\|_{\mathbf{M}_j^{-1}} - \|\mathbf{k}_j\|^2_{\mathbf{M}_j^{-1}} \|\mathbf{k}_j'\|^2_{\mathbf{M}_j^{-1}}$$

$$= 1 - \|\mathbf{k}_j\|^2_{\mathbf{M}_j^{-1}} - \|\mathbf{k}_j'\|^2_{\mathbf{M}_j^{-1}} + \lambda\left(1 - \|\mathbf{k}_j\|^2_{\mathbf{M}_j^{-1}}\right) + \lambda\left(1 - \|\mathbf{k}_j'\|^2_{\mathbf{M}_j^{-1}}\right) + \lambda^2$$

$$\qquad - k(\hat{\mathbf{x}}_j, \hat{\mathbf{x}}_j')^2 - 2k(\hat{\mathbf{x}}_j, \hat{\mathbf{x}}_j') \|\mathbf{k}_j\|_{\mathbf{M}_j^{-1}} \|\mathbf{k}_j'\|_{\mathbf{M}_j^{-1}}$$

$$\stackrel{(iii)}{\geq} 1 - \|\mathbf{k}_j\|^2_{\mathbf{M}_j^{-1}} - \|\mathbf{k}_j'\|^2_{\mathbf{M}_j^{-1}} + \lambda\left(1 - \|\mathbf{k}_j\|^2_{\mathbf{M}_j^{-1}}\right) + \lambda\left(1 - \|\mathbf{k}_j'\|^2_{\mathbf{M}_j^{-1}}\right) + \lambda^2$$

$$\qquad - k(\hat{\mathbf{x}}_j, \hat{\mathbf{x}}_j')^2 - 2k(\hat{\mathbf{x}}_j, \hat{\mathbf{x}}_j')$$

$$= \lambda^2 - 1 + (\lambda + 1)\left(1 - \|\mathbf{k}_j\|^2_{\mathbf{M}_j^{-1}}\right) + (\lambda + 1)\left(1 - \|\mathbf{k}_j'\|^2_{\mathbf{M}_j^{-1}}\right) - 2k(\hat{\mathbf{x}}_j, \hat{\mathbf{x}}_j') - k(\hat{\mathbf{x}}_j, \hat{\mathbf{x}}_j')^2$$

$$= \lambda^2 - 1 + (\lambda + 1)\left(\tilde{k}_j(\hat{\mathbf{x}}_j, \hat{\mathbf{x}}_j) + \tilde{k}_j(\hat{\mathbf{x}}_j', \hat{\mathbf{x}}_j')\right) - 2k(\hat{\mathbf{x}}_j, \hat{\mathbf{x}}_j') - k(\hat{\mathbf{x}}_j, \hat{\mathbf{x}}_j')^2$$

where $(i)$ follows from our assumption that $k(\mathbf{x}, \mathbf{x}) = 1$, $(ii)$ follows from the Cauchy-Schwarz inequality, and $(iii)$ follows since $1 - \|\mathbf{k}_j\|^2_{\mathbf{M}_j^{-1}} \leq 1$ and $1 - \|\mathbf{k}_j'\|^2_{\mathbf{M}_j^{-1}} \leq 1$.

Define $S_i := \sum_{\ell=1}^{i} N_i$ and $\tilde{v}_i := \min_{S_{i-1}+1 \leq j \leq S_i} \frac{1}{2}(\tilde{k}_j(\hat{\mathbf{x}}_j, \hat{\mathbf{x}}_j) + \tilde{k}_j(\hat{\mathbf{x}}_j', \hat{\mathbf{x}}_j'))$. Substituting this result into (9),

$$\sum_{t=1}^{r} \sigma_{t-1}(\mathbf{x}_t) \geq \lambda \Bigg( \sum_{j=1}^{N} \log \left( \lambda^2 - 1 + (\lambda + 1)\left(\tilde{k}_j(\hat{\mathbf{x}}_j, \hat{\mathbf{x}}_j) + \tilde{k}_j(\hat{\mathbf{x}}_j', \hat{\mathbf{x}}_j')\right) - 2k(\hat{\mathbf{x}}_j, \hat{\mathbf{x}}_j') - k(\hat{\mathbf{x}}_j, \hat{\mathbf{x}}_j')^2 \right)$$

$$+ \log \left| \lambda \mathbf{I} + \hat{\mathbf{K}}_{s-2N} \right| - 2 \Bigg)$$

$$= \lambda \Bigg( \sum_{i=1}^{m-1} \sum_{j=S_{i-1}+1}^{S_i} \log \left( \lambda^2 - 1 + (\lambda + 1)\left(\tilde{k}_j(\hat{\mathbf{x}}_j, \hat{\mathbf{x}}_j) + \tilde{k}_j(\hat{\mathbf{x}}_j', \hat{\mathbf{x}}_j')\right) - 2k(\hat{\mathbf{x}}_j, \hat{\mathbf{x}}_j')\right)$$

$$- k(\hat{\mathbf{x}}_j, \hat{\mathbf{x}}'_j)^2\Big) + \log\left|\lambda\mathbf{I} + \hat{\mathbf{K}}_{s-2N}\right| - 2\Bigg)$$

$$\geq \lambda\Bigg(\sum_{i=1}^{m-1}\sum_{j=S_{i-1}+1}^{S_i} \log\left(\lambda^2 - 1 + 2(\lambda+1)\tilde{v}_i - 2k(\hat{\mathbf{x}}_j, \hat{\mathbf{x}}'_j) - k(\hat{\mathbf{x}}_j, \hat{\mathbf{x}}'_j)^2\right)$$

$$+ \log\left|\lambda\mathbf{I} + \hat{\mathbf{K}}_{s-2N}\right| - 2\Bigg)$$

$$\overset{(i)}{\geq} \lambda\Bigg(\sum_{i=1}^{m-1}\sum_{j=S_{i-1}+1}^{S_i} \log\left(\lambda^2 - 1 + 2(\lambda+1)\tilde{v}_i - 2k\left(\sqrt{M_i}\right) - k\left(\sqrt{M_i}\right)^2\right)$$

$$+ \log\left|\lambda\mathbf{I} + \hat{\mathbf{K}}_{s-2N}\right| - 2\Bigg)$$

$$= \lambda\Bigg(\sum_{i=1}^{m-1} N_i \log\left(\lambda^2 - 1 + 2(\lambda+1)\tilde{v}_i - 2k\left(\sqrt{M_i}\right) - k\left(\sqrt{M_i}\right)^2\right)$$

$$+ \log\left|\lambda\mathbf{I} + \hat{\mathbf{K}}_{s-2N}\right| - 2\Bigg)$$

$$= \lambda\Bigg(\sum_{i=1}^{m-1} N_i \log\left(V_i - 2k\left(\sqrt{M_i}\right) - k\left(\sqrt{M_i}\right)^2\right) + W\Bigg) \tag{10}$$

$$=: \mathcal{L} \tag{11}$$

where $V_i := \lambda^2 - 1 + 2(\lambda+1)\tilde{v}_i$ and $W := \log\left|\lambda\mathbf{I} + \hat{\mathbf{K}}_{s-2N}\right| - 2$, $(i)$ follows from our assumption that the kernel $k$ is stationary and can be written in a single argument form as $k(\|\mathbf{x} - \mathbf{x}'\|) = k(\mathbf{x}, \mathbf{x}')$ and the fact that every pair of queries in $\mathcal{S}$ chosen with control set $i$ has squared distance at least $M_i$.

Starting from (4) in the proof of Theorem 4.1 except replacing the probabilities of all events with $2m/\delta$,

$$R_T \leq \left(\sum_{i=1}^{m} c_i(2\beta_T)\left(2\sum_{t\in\widetilde{T}_i}\sigma_{t-1}(\mathbf{x}_t) + 4\log\frac{2m}{\delta} + 8\log(4) + 1\right)\right) + c_m\sum_{t=1}^{T}\epsilon_t$$

$$= 2\beta_T\Bigg(c_m\left(2\sum_{t\in\widetilde{T}_m}\sigma_{t-1}(\mathbf{x}_t) + 4\log\frac{2m}{\delta} + 8\log(4) + 1\right)$$

$$+ \sum_{i=1}^{m-1} c_i\left(2\sum_{t\in\widetilde{T}_i}\sigma_{t-1}(\mathbf{x}_t) + 4\log\frac{2m}{\delta} + 8\log(4) + 1\right)\Bigg) + c_m\sum_{t=1}^{T}\epsilon_t$$

$$\overset{(i)}{\leq} 2\beta_T\Bigg(c_m\left(2\sum_{t\in\widetilde{T}_m}\sigma_{t-1}(\mathbf{x}_t) + 4\log\frac{2m}{\delta} + 8\log(4) + 1\right)$$

$$+ \sum_{i=1}^{m-1} c_i\left(2\sqrt{4(T_i+2)\gamma_{T_i}(\widetilde{\mathcal{X}_i})} + 4\log\frac{2m}{\delta} + 8\log(4) + 1\right)\Bigg) + c_m\sum_{t=1}^{T}\epsilon_t$$

$$\overset{(ii)}{\leq} 2\beta_T\Bigg(c_m\left(2\sqrt{4(T+2)\gamma_T(\mathcal{X})} - \sum_{t=1}^{r}\sigma_{t-1}(\mathbf{x}_t) + 4\log\frac{2m}{\delta} + 8\log(4) + 1\right)$$

$$+ \sum_{i=1}^{m-1} c_i\left(2\sqrt{4(T_i+2)\gamma_{T_i}(\widetilde{\mathcal{X}_i})} + 4\log\frac{2m}{\delta} + 8\log(4) + 1\right)\Bigg) + c_m\sum_{t=1}^{T}\epsilon_t$$

$$\overset{(iii)}{\leq} 2\beta_T\Bigg(c_m\left(2\sqrt{4(T+2)\gamma_T(\mathcal{X})} - \mathcal{L} + 4\log\frac{2m}{\delta} + 8\log(4) + 1\right)$$

$$+ \sum_{i=1}^{m-1} c_i \left( 2\sqrt{4(T_i+2)\gamma_{T_i}(\widetilde{\mathcal{X}}_i)} + 4\log\frac{2m}{\delta} + 8\log(4) + 1 \right) \right) + c_m \sum_{t=1}^{T} \epsilon_t$$

$$= \mathcal{O}\Bigg( \left( B + \sqrt{\gamma_T(\mathcal{X}) + \log\frac{2m}{\delta}} \right) \left( c_m \left( \sqrt{T\gamma_T(\mathcal{X})} - \mathcal{L} + \log\frac{2m}{\delta} \right) \right.$$

$$\left. + \sum_{i=1}^{m-1} c_i \left( \sqrt{T_i\gamma_{T_i}(\widetilde{\mathcal{X}}_i)} + \log\frac{2m}{\delta} \right) \right) \Bigg) + c_m \sum_{t=1}^{T} \epsilon_t$$

where $(i)$ follows from Lemma A.3, $(ii)$ follows from Lemma A.3 again and the resulting inequality $\sum_{t=1}^{r} \sigma_{t-1}(\mathbf{x}_t) + \sum_{t=r+1}^{T} \sigma_{t-1}(\mathbf{x}_t) \leq \sqrt{4(T+2)\gamma_T(\mathcal{X})}$, and $(iii)$ follows from substituting in (11). A union bound over the events of the single application of Lemma A.4, $m$ applications of Lemma A.5, and $m-1$ applications of Lemma 4.3 yields the desired $1 - \delta$ probability bound, which completes the proof. $\qquad\square$

## A.6 Other Lemmas

**Lemma A.2** ([7] Lemma 3). *Let $(\mathbf{x}_t)_{t=1}^{T}$ be a sequence of queries selected by some algorithm. Then, the mutual information $I(\mathbf{y}_{1:T}; \mathbf{f}_{1:T})$ between the noisy observations $\mathbf{y}_{1:T}$ and the function values $\mathbf{f}_{1:T}$ at the queries is given by*

$$I(\mathbf{y}_{1:T}; \mathbf{f}_{1:T}) = \frac{1}{2}\log\left|\mathbf{I} + \lambda^{-1}\mathbf{K}_t\right| = \frac{1}{2}\sum_{t=1}^{T}\log(1 + \lambda^{-1}\sigma_{t-1}^2(\mathbf{x}_t)).$$

**Lemma A.3** ([7] Lemma 4). *Let $(\mathbf{x}_t)_{t=1}^{T}$ be a sequence of queries selected by some algorithm. Then*

$$\sum_{t=1}^{T} \sigma_{t-1}(\mathbf{x}_t) \leq \sqrt{4(T+2)\gamma_T(\mathcal{X})}\,.$$

**Lemma A.4** ([7] Theorem 2). *Let $\beta_t := B + \sigma\sqrt{2\left(\gamma_{t-1}(\mathcal{X}) + 1 + \log(1/\delta)\right)}$ where $B$ is the upper bound of the RKHS norm of $f$. With probability at least $1 - \delta$, for all $\mathbf{x} \in \mathcal{X}$ and $t \geq 1$,*

$$|\mu_{t-1}(\mathbf{x}) - f(\mathbf{x})| \leq \beta_t \sigma_{t-1}(\mathbf{x})$$

*where $\mu_{t-1}$ and $\sigma_{t-1}$ are defined in (1) with $\lambda = 1 + \eta$ and $\eta := 2/T$.*

**Lemma A.5** ([16] Lemma 3). *Let $X_t$ be any non-negative stochastic process adapted to a filtration $\{\mathcal{F}_t\}$, and define $m_t := \mathbb{E}[X_t|\mathcal{F}_{t-1}]$. Further assume that $X_t \leq b_t$ for a fixed, non-decreasing sequence $(b_t)_{t\geq 1}$. If $b_T \geq 1$, with probability at least $1 - \delta$, for any $T \geq 1$,*

$$\sum_{t=1}^{T} m_t \leq 2\sum_{t=1}^{T} X_t + 4b_T\log\frac{1}{\delta} + 8b_T\log(4b_T) + 1$$

# B  Comparison to Naive Baselines

We investigated simple extensions of TS-PSQ, UCB-PSQ, and EI-PSQ (i.e., the classic expected improvement algorithm [20] adapted for BOPSQ, see Appendix C for details) for the cost-varying problem by dividing the acquisition score of a control set by its cost in a manner similar to Snoek et al. [31, Sec. 3.2]. Fig. 3 shows the mean and standard error of the simple regret incurred over 10 RNG seeds for one set of experiments. We found that these naive methods generally do not work well. For TS per unit cost and UCB-PSQ per unit cost, if a suboptimal control set is very cheap, its acquisition score may remain artificially high throughout, and the algorithm fails to converge. EI per unit cost was slightly more promising, but suffered from the inverse problem: the suboptimal control sets had 0 expected improvement very quickly and dividing by the cost had no effect. This algorithm thus fails to encourage exploration with cheaper control sets. Furthermore, the EI algorithm was computationally expensive due to the double Monte Carlo expectation computation. In general, we see that the UCB-CVS algorithm is able to use the cheaper control sets much more effectively for exploration and hence find better solutions.

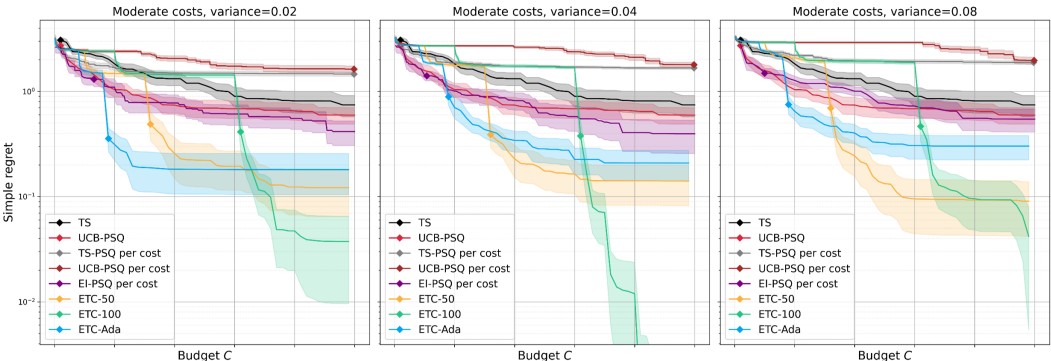

Figure 3: Mean and standard error (over 10 RNG seeds) of the simple regret (lower is better) incurred against cost spent (budget) $C$ by all algorithms including TS-PSQ per unit cost, UCB-PSQ per unit cost, and EI per unit cost, with samples from the GP prior as the objective function, moderate cost set, and all variances. A diamond indicates the average budget after which an algorithm only chooses the optimal control set.

# C  Experimental Details

All experiments use a squared exponential kernel with ARD lengthscales that depend on the objective function, $k(\mathbf{x}, \mathbf{x}') = 1$, Gaussian observation noise with $\sigma = 0.01$, and 5 initial query-observation pairs with queries drawn uniformly at random. All expectations are approximated with Monte Carlo sampling with 1024 samples. All acquisition maximizations are performed with L-BFGS-B with random restarts. All query sets are $[0, 1]^d$.

## C.1  Objective functions

The control sets described here are given in an order corresponding to their costs given in Sec. 5. For example, for the GP samples objective, under the cheap cost set, control set $\{1\}$ has cost $0.01$, control set $\{1, 2\}$ has cost $0.1$, and control set $\{1, 2, 3\}$ has cost 1.

**Samples from GP prior (3-D):** We use samples from the same kernel $k$ used to model the GP posteriors during learning. We use a kernel lengthscale of $0.1$ and control sets $\{\{1\}, \{2\}, \{3\}, \{1, 2\}, \{1, 3\}, \{2, 3\}, \{1, 2, 3\}\}$.

**Hartmann (3-D):** We use a kernel lengthscale of $0.1$ and control sets $\{\{1\}, \{2\}, \{3\}, \{1, 2\}, \{1, 3\}, \{2, 3\}, \{1, 2, 3\}\}$.

**Plant growth simulator (5-D):** The plant growth simulator is a GP built from private data collected on the maximum leaf area achieved by Marchantia plants depending on input variables Ca, B, NH$_3$, K, and pH. We use min-max feature scaling to scale all input variables to $[0, 1]$ and standardize the output values. We use the posterior mean of the GP as the objective function. We use a kernel lengthscale of $0.2$ and control sets $\{\{1, 2\}, \{3, 4\}, \{4, 5\}, \{1, 2, 3\}, \{2, 3, 4\}, \{3, 4, 5\}, \{1, 2, 3, 4, 5\}\}$.

**Airfoil self-noise (5-D):** We use the airfoil self-noise dataset from the UCI Machine Learning Repository [9]. To scale all input variables to $[0, 1]$, we first take the natural logarithm of variables 1 and 5, then do min-max feature scaling on all input variables. We also standardize the output values. We then feed the data into a default SingleTaskGP from BoTorch and use the posterior mean as the objective function. We use a kernel lengthscale of $0.2$ and control sets $\{\{4, 5\}, \{2, 5\}, \{1, 4\}, \{2, 3\}, \{3, 5\}, \{1, 2\}, \{3, 4\}\}$.

## C.2  Algorithms

**UCB-PSQ and UCB-CVS:** For the experiments, we set $\beta_t = 2$ for all $t$.

**TS-PSQ:** Following [13], we use random Fourier features (RFF) [28] to approximately sample from a GP posterior. We use RFF with 1024 features.

**EI-PSQ:** We adapt the BoTorch acquisition NoisyExpectedImprovement to the BOPSQ problem setting. To evaluate the acquisition score of a partial query, we first sample 32 fantasy models of $f$ from the GP posterior. For each fantasy model, we compute the expected value of the partial query and take the best value as the value of the best observation so far (assuming the full query control set is available). We then compute the improvement score as the expected value minus the best value, and then average the improvement score over all fantasy models.

### C.3 Implementation

The experiments were implemented in Python. The major libraries used were NumPy [12], SciPy [40], PyTorch [26], GPyTorch [10] and BoTorch [3]. For more details, please refer to the code repository.

### C.4 Compute

The following CPU times in seconds were collected on a server running Ubuntu 20.04.4 LTS with $2\times$ Intel(R) Xeon(R) Gold 6326 CPU @ 2.90GHz and 256 GB of RAM. We measure the CPU time for 1 iteration of TS-PSQ and UCB-CVS with a dataset of 100 observations. In general, none of the algorithms in the settings tested in this paper require a significant amount of compute.

|         | GP sample | Hartmann | Plant | Airfoil |
|---------|-----------|----------|-------|---------|
| TS-PSQ  | 6.27      | 4.14     | 8.96  | 232.27  |
| UCB-CVS | 37.92     | 52.34    | 61.96 | 87.09   |

## D  Limitations

A limitation of our work is that the theoretical guarantees of UCB-CVS rely on a few assumptions that may not hold in practice. For example, the regularity assumption that assumes the objective function $f$ resides in some RKHS may not be true in some problems. The kernel corresponding to this RKHS may not be known either. The work also assumes that the probability distributions governing each variable are independent and fixed. In practice, these assumptions may be violated, if the probability distributions have some dependence on one another, or may change over time.

