# OpenReview forum: "Bayesian Optimization with Cost-varying Variable Subsets"
_NeurIPS.cc/2023/Conference — NeurIPS 2023 poster_

### Official Review · Reviewer_o8eg · 2023-06-29

**Soundness:** 3 good
**Presentation:** 2 fair
**Contribution:** 2 fair
**Rating:** 5
**Confidence:** 3

**Summary:**

The paper introduces a novel Bayesian Optimization algorithm for the case where we select a subset of variables to query at each iteration. Furthermore, each subset will incur a different _cost_. Examples of this include: control of soil nutrients in farming, advanced manufacturing, and targeting specific subgroups for ad revenue. Previous work has focused on optimizing across subsets, but does not consider the cost. Due to the method being based on Thompson Sampling, it is not simple to extend to the cost-variable setting.

The problem setting considers the case where we can choose a subset of inputs (a control set) and its corresponding values, then the remaining inputs are samples randomly from a _known_ distribution. This setting differs from multi-fidelity problems because lack of information comes from randomness in the choice of inputs when they are not selected as part of the control set, not from querying a cheaper approximation. Our objective is to find the control set and its corresponding values that maximize the expected value of the function (with respect to the randomness in the remaining inputs).

The algorithm is based on the upper confidence bound (UCB) acquisition function, and mainly works in four stages:

(a) We find the maximum expected UCB across all valid input subsets

(b) We build a set of possible control sets that have a maximum value $\epsilon_t$ close to the maximum from (a)

(c) We build a set of that includes all the control sets with minimal cost from (b)

(d) Finally, from the remaining possible control sets, we choose the set with the maximum expected UCB to query

Note that an important hyper-parameter sequence, $\{ \epsilon_t \}$ as been introduced. Then four theoretical results are presented. Theorem 4.1 shows that the algorithm is no-regret for appropriately chosen $\beta_t$ and a sub-linear $\sum_{t = 1}^T \epsilon_t$. Proposition 4.2 shows that provides an alternative condition for which the no-regret property hold. Then Lemma 4.3 and Theorem 4.4 show that the bound depends on the variance of the input distributions, with larger variance allowing for more efficient exploration. They further show that by choosing an $\epsilon$-schedule that increases the number of times cheaper controls are played, we can make the bound tighter. The $\epsilon$-schedule was useful for theoretical analysis, but in practice it is dropped instead for a simpler idea, which focuses on fixing the number of plays for each cost-group, choosing the lowest possible cost group that still has plays at each iteration when all cost groups have been explored, the algorithm reverts to the standard version with $\epsilon_t = 0$.

Experiments are carried out in four benchmarks: a 3D GP sample, Hartmann3D, a plant growth simulator, and a simulator built from the airfoil self-noise data-set. Experiment show that when there are subsets with small or moderate costs, the algorithms give an advantage to the Thompson Sampling baseline. When costs are expensive, algorithm performance drops when cost-group sizes are fixed, but if made adaptive the performance is still competitive. In addition, the simple baselines perform well when the control sets are not subsets of each other (and therefore they are not constantly querying the most expensive one). Finally, the results show that having a larger variance for the random inputs tends to give better regret, but also appears to be problem and algorithm dependent.

**Strengths:**

Originality: The author propose an algorithm that tackles a problem not fully considered in Bayesian Optimization, where there are subsets of inputs that can be queried and each subset has a different cost. They also include a thorough theoretical analysis of the regret of the algorithm, and show empirical evidence of its effectiveness. Although I have not seen this specific problem solved before, there are other works could perhaps tackle the problem (see more later).

Quality: The algorithm proposed is sound, combining known Bayesian Optimization ideas that are known to work well. The theoretical analysis is very good and complete, exploring the effect of different aspects of the problem (e.g. $\epsilon$-schedule and variance). The empirical analysis is perhaps limited to a few examples, but they provide a varied number of costs and variances which is important.

Clarity: Generally the paper is very clear, and well written. Important equations are well explained, and the implications of all theoretical results made clear. Figure 1 is very good at explaining the problem setting, and helping understand notation. Empirical results are explained and analyzed thoroughly and clearly. Perhaps a few paragraphs of the paper could be rewritten, and result exposition improved (see later).

Significance: The problem is significant to the wider scientific community, and relevant examples are given. Although it seems limited by certain assumptions and perhaps giving more detail on the examples would be better (see later).

**Weaknesses:**

- The paper seems highly relevant to Causal Bayesian Optimization [1, 2, 3], where they (a) choose a subset of inputs and the rest are observed from the environment, and (b) they consider the cost of the varying subsets. In particular, they do not assume that the input distributions are known but that they instead follow an unknown causal structure which appears to be a more general of a case. Indeed it could be argued that all motivating examples mentioned by the paper fall under this umbrella (i.e. there is an underlying causal structure). Empirical comparison against their baselines would be very strong, but at least a discussion of where the settings differ should be included.

- Following to the previous point, the assumption that the input distributions are known and independent of the chosen values for the control set appears to be very restrictive and limits the application of UCB-CVS. I assume the distribution could be estimated from data, so perhaps there are extensions for the proposed method when this is the case. Based on this, the significance of the proposed paper appears to be far more restrictive than claimed as it only applies when the distributions are known.

- The theoretical analysis is perhaps the strongest part of the paper, however in the end the authors opted for for a rather arbitrary way of selecting which subsets to query (i.e. by defining cost groups and selecting an allocation for each). The link between the theoretical results and the new heuristic should be made clearer, in addition, the heuristic appears to me to be sub-optimal especially in the case where an expensive subset leads to clearly sub-optimal observations but we keep querying it anyway due to the requirement to query each cost group a certain number of times.

Minor weaknesses:

- I found the first paragraphs of section 4 (lines 115 to 138) to be a little convoluted and difficult to read. I understand there is a lot of notation to introduce, but perhaps it would benefit from a small re-write. Figure 1 does make an excellent job of summarizing it all and was very helpful.

- I found the plots in Figure 2 to be a little difficult to read, the plots are small and pixelated, and the font sizes could be made bigger (there seems to be a lot of repetition in the titles so perhaps things could be condensed to use less words and make space for larger fonts).

- Only using 10 runs per example seems a little bit low, would be better to use more.

[1] Aglietti, Virginia, et al. "Causal bayesian optimization." International Conference on Artificial Intelligence and Statistics. PMLR, 2020.

[2] Branchini, Nicola, et al. "Causal entropy optimization." International Conference on Artificial Intelligence and Statistics. PMLR, 2023.

[3] Tigas, Panagiotis, et al. "Differentiable Multi-Target Causal Bayesian Experimental Design." ICML, 2023.

**Questions:**

- I saw in the appendix that you included some obvious naive baselines, in particular EI-PSQ-per-cost seemed to give stronger results than the baselines compared against in the main paper. Why was it not included?

- We are seeking to maximize the expected value of the function under randomness from the non-control inputs, can you imagine a scenario where low variance would also be important? If so, would would the method be compatible with a simple scalarazation (ie. $i^*, x^* = \arg\max \left[ E(f(x)) + \alpha Var(f(x)) \right]) $ to solve the problem?

- The algorithm's performance seems to be very sensible to the choice of the $\epsilon$-schedule. Is there any more guidance on how to choose it? Indeed, the ETC variants appear to perform very differently depending on the task.

**Limitations:**

Limitations are addressed in the appendix: the theoretical assumptions may not hold in practice, and the assumption that the input probability distributions are independent and fixed may be violated (this is considered by Causal BO). The authors also mention poor scaling when the number of control sets is large.

---

> ### Author Rebuttal · Authors · 2023-08-04
>
> Thank you for your time and effort spent writing this highly detailed and insightful review. Allow us to answer some your concerns:
>
> **Weakness 1: relation to Causal BO.** You are correct in pointing out that Causal BO as formulated in Aglietti et al. (2020) is a more general case. Specifically, our setting is a case in which there are no 'non-manipulative' variables and the causal DAG is such that all input variables have no parents and are parents of the output variable. Nevertheless, we believe our focus on this special case has value as it allows us to derive useful theoretical results that do not exist thus far for the completely general setting. To the best of our knowledge, the three works you cited and many other Causal BO works do not have regret bounds, with the exception of "Model-based Causal Bayesian Optimization" (Sussex et al., 2023). This work is also a special case of Aglietti et al. (2020), and has little overlap with our work as it does not consider costs of control sets or explicit probability distributions over input variables. This work and our work are products of the general principle that focusing on special cases enables the derivation of non-trivial theoretical results which may be very difficult to derive for the general case (supported by the lack of theory for the general case thus far). We believe that our work is sufficiently general to be useful for practical scenarios (where the full Causal BO apparatus may be unnecessary), and is also a stepping stone towards theory for the general case. We agree with you that the relation of our work to the Causal BO literature should be discussed and will certainly include this discussion in the final paper.
>
> **Weakness 2: assumption of known distributions.** We think this assumption is a reasonable one for three reasons:
> 1. There are many scenarios in real life in which the distribution of some random variable of interest is known or at least can be estimated accurately from historical data. For instance, in our example of ad revenue maximization or crowdsourcing, where the query variables describe human demographics such as country of origin or income level, the platform being used would have access to a large amount of user demographics data.
> 2. This is a standard assumption made in this line of work due to the difficulty of saying or doing anything useful with too little assumptions. Many prior BO works that cater to uncontrollable random variables make this or similar assumptions; for instance, the cited Causal BO work (Aglietti et al., 2020) assumes the presence of a dataset from an observational study. Other works that make the assumption of known distributions include BO for expected values (Toscano-Palmarin & Frazier, 2022), risk-averse BO (Cakmak et al., 2020; Nguyen et al., 2021a;b), and distributionally robust BO (Kirschner et al., 2020; Nguyen et al., 2020; Tay et al., 2022).
> 3. As you have intuited, it is possible to modify the setting to one in which the distributions are not known _a priori_ but are estimated in the course of the optimization process. Previous BO works that have done this include the 'data-driven' setting  (Hayashi et al., 2022; Kirschner et al., 2020). Since the extension has already been developed in the literature, accounting for this procedure in our algorithm and theory within the constraints of a conference paper incurs additional complications that distract from our work's primary conclusions for little novelty. We will include a discussion on this extension along with references to these previous works in the final paper.
>
> **Weakness 3: choice of heuristic.** The heuristic was chosen based on the interpretation of Theorem 4.4 that an $𝜖$-schedule that increases the number of times cheaper control sets are played can reduce the MIG term of the most expensive control set $m$, which implies that if $c_i≪c_m$ for all $i$, we can reduce overall regret by playing cheaper control sets a greater number of times. The ETC heuristic is a method of enforcing these plays, whereas trying to define an $𝜖$-schedule may result in cheaper control sets not being played at all if the chosen values of $𝜖$ are too small, and knowing the range of good values of $𝜖$ _a priori_ is impossible since $f$ is unknown. We agree with you that the heuristic (with a fixed no. of plays for each control set) is sub-optimal if there is an expensive sub-optimal control set, and Theorem 4.4 reflects that as well: taking $m$ to be optimal, if there is a sub-optimal $i$   such that $c_i ≈ c_m$, then the decrease in the overall regret is not as large as if $c_i≪c_m$ for all $i$. To overcome this, instead of a fixed number of plays for each control set, we suggest in the paper to choose the number of plays adaptive to the cost, and our ETC-Ada variant in the experiments suggests that an $\mathcal O(c_i^{-1})$ number of plays for each control set $i$ generally performs well across many different settings and is a robust choice for practitioners. We will make this justification clearer in the final paper.
>
> **Questions:**
> 1. We did not plan to include these obvious naive baselines in the main paper as we hypothesized that they would not work well due to the reasons explained in Appendix B, and they were not required to arrive at our main conclusions in the Experiments sections. We agree it would be interesting to see how EI-PSQ-per-cost performs in all settings. We will run this baseline in all settings and include the results in the main body in the final paper.
> 2. Certainly, there have been many BO works that aim to maximize some risk-sensitive objective such as mean-variance as you mentioned (Iwazaki et al., 2021), or others such as VaR/CVaR (Nguyen et al., 2021a;b). We suspect it will not be too difficult to replace the expected value in our proposed method with these alternate objectives.
> 3. See discussion on ETC-Ada in Weakness 3.
>
> We hope the clarifications above can improve your opinion of our work.

---

> > ### Comment · Reviewer_o8eg · 2023-08-10
> > **Response to Rebuttal**
> >
> > Thank you for the detailed response. While I still believe that the paper would benefit from empirical evaluation to Causal BO, I agree that (i) the current setting allows for theoretical analysis and this should not be understated and (ii) a discussion of the the Causal BO should be at least included. I have also been convinced that the assumption of known distributions, while limiting, is perhaps reasonable in many applications. The link between theory and the choice of heuristic is clearer to me now, and while the sensitivity to the choice of $\epsilon$-schedule remains, I agree that the adaptive variant provides a somewhat robust option.
> >
> > I seem unable to update my score at the current time, but based on the above, I will upgrade my recommendation by one point when possible.

---

### Official Review · Reviewer_xcAB · 2023-07-05

**Soundness:** 3 good
**Presentation:** 3 good
**Contribution:** 3 good
**Rating:** 7
**Confidence:** 4

**Summary:**

The submission studies the problem of Bayesian optimization (BayesOpt) where we can choose to control a subset of the decision vector while the other variables are randomly selected.
Unlike previous works on contextual BayesOpt and BayesOpt with uncertain input, this paper leaves the choice of which variables are set to the user, and accounts for the associated cost of controlling a given set of variables.
As far as I know, this is the first work to tackle this problem.
The authors then proposed an Upper Confidence Bound (UCB) style algorithm that seeks the subset of variables that yields the highest expected upper bound on the objective function's value while incurring the lowest cost.
Theoretical guarantees are derived for the proposed algorithm, showing the algorithm achieves sub-linear cumulative regret.
Numerical experiments show that variants of the proposed algorithm perform competitively against two baseline policies.

**Strengths:**

The submission tackles an original problem that is motivated by many real-world scenarios.
The paper is well-written, and the exposition is clear.
The theoretical results are explained well and are shown to be helpful in interpreting results from numerical experiments.
Numerical experiments on the other hand show strong performance from the proposed algorithm.

**Weaknesses:**

I don't have many complaints about the paper.
The authors can consider including the other baselines shown in Appendix B in Sec. 5 instead of simply referring to them.
The authors can also consider including a simple baseline that randomly selects the subset of variables to control for at each iteration.

**Questions:**

- The work assumes that the uncontrolled variables are independently sampled from their respective distributions, but I imagine there are scenarios in which the decision variables aren't conditionally independent. Could the proposed approach tackle this scenario too? Perhaps the correlations between known dependent variables could be modeled somehow.
- My understanding is that we assume the costs $c_i$ associated with controlling given subsets are known _a priori_. Do the authors have any comment on how much relaxing this assumption would complicate what we have? I imagine Line 5 in Algorithm 1 might be harder to implement.
- How was $4$ chosen in the **ETC-Ada** variant's $4/c_j$ threshold? Should this be the same constant in all settings?

**Limitations:**

The authors discuss the difficulty of setting the $\varepsilon$-schedule in Sec. 4.3 and recommend an explore-then-commit strategy.

---

> ### Author Rebuttal · Authors · 2023-08-07
>
> Thank you for the time and effort spent writing this review. We answer your questions below:
>
> **Questions**:
> 1. Certainly, as long as the joint distributions are known and induce conditional expectations that can be computed or approximated via Monte Carlo sampling. Algorithm 1 will likely work well under such conditions. The only parts that will be affected are the theoretical results Lemma 4.3 and Theorem 4.4 as they assume independence in order to relate the variance of the individual distributions governing each variable to the regret bound. Some more involved analysis will be required to say something useful in the general case when the distributions may not be independent.
> 2. You are correct that Algorithm 1 cannot be used effectively if the costs are not known. For example, Algorithm 1 would never play a control set $i$ if there was another control set $j$ that was both cheaper (i.e., $c_j < c_i$) and had a higher expected UCB score (i.e., $\mathbb E[u_{t-1}([\mathbf x^i, \mathbf X^{-i}])] < \mathbb E[u_{t-1}([\mathbf x^j, \mathbf X^{-j}])] $). Without knowing the costs, such comparisons between control sets cannot be made. Fortunately, this assumption of known costs is mild as they are problem-specific constants usually known in practical scenarios, or at least can be found with little difficulty.
> 3. $4$ was an arbitrarily chosen constant. In general, we wished to convey that an $\mathcal O(c_j^{-1})$ threshold is likely to work well across different sets of costs and is a robust choice for practitioners that keeps the number of hyperparameters to a minimum. We will make this clearer in the final version of our paper.
>
> If you have any further questions, please let us know and we will be happy to answer them.

---

> > ### Comment · Reviewer_xcAB · 2023-08-10
> > **thanks**
> >
> > I thank the authors for their response. I will keep my score as is, though I do have two quick notes:
> > - I would not make the (in my opinion strong) claim that having known costs is a mild assumption or costs can be determined easily. In many applications, we might be faced with a very complex cost surface that's not easily learned. However, I'm completely fine with the method being able to handle only scenarios with known costs, as I do agree many applications fall under this setting.
> > - Interesting point about the constant $4$. Did you try other constants? If not, it might be good to include some discussion on this.

---

### Official Review · Reviewer_aGSh · 2023-07-06

**Soundness:** 4 excellent
**Presentation:** 3 good
**Contribution:** 3 good
**Rating:** 6
**Confidence:** 3

**Summary:**

This work introduces a new black-box function optimization setting where only a collection of the subsets of variables can be optimized while the values of the complement variables for each set are randomly sampled. The authors propose a new BO framework based on GP-UCB, called UCB-CVS, to solve this optimization setting.

**Strengths:**

1.	The problem setting is new and hasn’t been fully investigated before.
2.	The algorithm proposed in this work is clearly described and the authors also provide theoretical guarantees.
3.	Experimental results show good performance of this new method.


**Weaknesses:**

1.	The extent of the problem setting's relevance and interest to the Bayesian Optimization (BO) community is not apparent to me in this work, despite its novelty. The authors evaluate their methods using two real-world datasets, namely plant growth and airfoil self-noise. However, the control sets, cost values, and probability distributions are still simulated by the authors themselves. Consequently, they have not demonstrated the application of real-world problems that can be effectively addressed within this setting using the new UCB-CVS algorithm.
2.	It is not so convincing to me that the terms in the cumulative regret $R_{T}$ should be multiplied by the cost $c_{i_t}$. The objective function presented in line 138 does not incorporate the cost coefficient, and it seems natural for the cumulative regret $R_{T}$ to also exclude the cost coefficient. Moreover, this approach would align better with the single regret mentioned in line 190. The cost information is already encompassed within the value of $T$, namely, for a fixed budget $C$, if we consistently select high-cost sets, $T$ would be small, resulting in insufficient observations. Conversely, if we consistently choose low-cost sets, although $T$ would be large, the optimization procedure is more likely to converge to sub-optimal solutions. Therefore, there is no necessity to introduce an additional cost coefficient for penalization purposes.
3.	The font size of Figure 2 is too small to be read.


**Questions:**

1.	UCB formula below line 161, why the subscript of $u$, $\mu$ and $\sigma$ is $t-1$ while $\beta$ is $t$?
2.	Proposition 4.2, is this result true with high probability or always true?
3.	Equation (2), is $M_i$ here the same as $M_i$ in Lemma 4.3?


**Limitations:**

The authors don't specifically discuss the limitations of this work. The authors can add a paragraph in their manuscript to discuss the limitations of their work based on the reviewers' feedback.

No significant negative societal impact of this work.

---

> ### Author Rebuttal · Authors · 2023-08-07
>
> Thank you for taking the time to read and review our paper. Let us address some of your concerns:
>
> **Weakness 1, on application to real-world problems**:  We have attempted to demonstrate the general applicability of our algorithm by using multiple cost sets. As explained in Sec. 5, while these cost sets may (at first glance) seem arbitrary, it is the algorithms’ relative performance across these cost sets rather than the absolute performance on a single cost set that allows us to understand the conditions under which particular algorithms perform better or worse. Real-world problems will come with their own cost sets defined by real-world constraints. If the real costs can also be categorized in a similar relative way like the above cheap, moderate, and expensive cost sets, then the results are expected to be similar.
>
> Unfortunately, we do not have the means (e.g., equipment, budget, domain experts, etc) to conduct real-world experiments and obtain these problem-dependent quantities that correspond exactly to some real-world problem. We hope you will understand the limitations we work with.
>
> **Weakness 2, on cost term in cumulative regret**: We believe that this is a difference of perspective: our perspective of the cumulative regret $R_T$ is that an algorithm should try to minimize the bound on $R_T$ for any given value of $T$, which is the original definition of regret in conventional BO that we have tried to adhere to for familiarity. Your redefinition prescribes the following: an algorithm should try to minimize the bound on the average regret $R_T/T$ for any given value of budget $C$. Since the average regret is an upper bound on the simple regret $\min_{t\leq T} r_t$, this definition is perfectly valid as well, though the analysis required to reach the conclusions of this work regarding the impact of different costs may be significantly different from ours and may be more difficult. We believe that our regret notion is more amenable to analysis. Furthermore, even though the objective below line 138 does not incorporate costs, the maximizer (i.e., $\mathbf x^{i_t}$ s.t. $r_t = 0$) coincides for both our regret notion and your proposed regret notion, hence our regret notion is also a suitable choice given that objective.
>
> We opted to make the cost terms explicit in $R_T$ also in order to avoid misinterpretations such as the following:  suppose that 1) the regret incurred in a single iteration were defined as $r_t = \mathbb E[{f([\mathbf x^{i^*}, \mathbf X^{-i^*}])}] -\mathbb E[{f([\mathbf x^{i_t}, \mathbf X^{-i_t}])}]$ without the cost term; 2) there are two control sets $j$ and $k$ such that $j$ is much cheaper than $k$, i.e., $c_j \ll c_k$; and 3) that the learner is in the initial exploration stage in the BO procedure such that the partial queries $\mathbf x^{j}$ and $\mathbf x^{k}$ at each iteration are more or less random and $\mathbb E[{f([\mathbf x^{j}, \mathbf X^{-j}])}] \approx  \mathbb E[{f([\mathbf x^{k}, \mathbf X^{-k}])}]$. Then, under the above redefinition of regret, the cumulative regret incurred in this exploration stage is the same whether the learner exclusively used control set $j$ or control set $k$. However, clearly, the learner is well-advised to use $j$ for exploration rather than $k$. Under the traditional interpretation that $r_t$ is a measure of how 'bad' the decision taken at iteration $t$ is, making the cost terms explicit naturally incorporates the notion that the penalty for sub-optimal plays is lower if the play was cheap, while also penalizing using the entire budget on sub-optimal plays, regardless of whether those plays are cheap or expensive.
>
> **Questions**:
> 1. We adopt the notation from Chowdhury and Gopalan (2017). The reasoning is that, at iteration $t$, before deciding the control set and partial query, the learner only has access to the dataset of observations up to time $t-1$. $\mu_{t-1}$ and $\sigma_{t-1}$ are the GP posterior mean and standard deviation given the dataset up to time $t-1$. The sequence $\\{\beta_t\\}_{t=1}^T$ is an algorithm parameter and is fully known at all iterations.
> 2. Thank you for pointing out this omission, the result is true with high probability. We will correct the proposition to include this in the final paper.
> 3. Yes, there may be some confusion here as Equation (2) is part of Lemma 4.3. Equation (2) puts a lower bound on $M_i$ that depends on the variances of the probability distributions involved.
>
> **Limitations**:
> We have a section discussing limitations in the Appendix.
>
> We hope our response above has adequately addressed your concerns. Please reach out to us if you have any further questions.

---

> > ### Comment · Reviewer_aGSh · 2023-08-15
> >
> > I want to thank the reviewers for their response. I have some follow-up comments.
> >
> > 1.
> >
> > I want to clarify that for real-world problems, it does not mean that the authors should do some experiments in the real world. However, ideally, the authors should give some explanations of why the simulated real-world problems are close to the real-world settings. As the authors say in their manuscript, "If the real costs can also be categorized in a similar relative way like the above cheap, moderate, and expensive cost sets, then the results are expected to be similar.", indicating that the authors don't clearly know whether the cost settings (as well as other settings such as the number of control sets, the probability distribution) is close to the real-world settings.
> >
> > However, I also admit that "why this problem is important?" is always a question that will not have a standard answer. Therefore, as reflected in the score, overall I have a positive impression of this work. I also want to know what other reviewers think of this point, as I notice, not just I have some concerns about the real-world settings in the manuscript.
> >
> > 2.
> >
> > If I understand correctly, the authors point out that one difference between my definition of regret and the authors' definition of regret is that the authors' definition does not have a fixed budget $C$ constraint. If so, then I don't see why, in the case that the authors construct, we want to use $j$ rather $k$. If we don't have a budget constraint, it doesn't matter whether we use a cheaper set or a more expensive set.  I think the reason that the authors need to have cost coefficients in the cumulative regret is just that the cost constraint is missing from the definition, therefore, another way should be done to incorporate this information into the regret.
> >
> > Besides, when we talk about the cumulative regret, usually we care about the sublinear regret, meaning that whether $R_{T}/T$ goes to $0$ when $T$ goes to infinity. Therefore, in that constructed case, in the initial exploration stage, in some way, I don't think $k$ is "worse" than $j$ because the choice will not affect the convergence behavior.
> >
> > While I am not very convinced that the authors' definition is more natural and closer to the original definition of regret in conventional BO, I agree that their regret notion is more amenable to analysis and it is still an interesting setting.

---

> > > ### Comment · Reviewer_o8eg · 2023-08-15
> > >
> > > Answering reviewer aGSh's feedback on (1.):
> > >
> > > I share concerns with the reviewer about the applicability of the algorithm. In particular about independence assumption for the inputs as this seems very restrictive (as mentioned in my review, I think Causal BO is a better fit for all problems mentioned in the paper). The author's simulating the probability distributions themselves acts in the benefit of their algorithm, and it remains unclear wether distributions would be similar in the real world. On the other hand, they make the independence assumption clear and this is where they display results (as well as mentioning the assumption as a limitation).
> > >
> > > However, regarding cost sets I believe the authors did a good job at exploring a large range of them, and I think this is more illuminating than trying just one cost set that is aligned with the real-world. They show good performance when the most expensive set is 100 times the cost of the cheapest one, when the most expensive is 10 times the cost, and when the most expensive is 1.67 times the cost. I agree with the authors that the performance across cost-sets is very important and it would be likely that real world costs would fall in a similar category to either 'cheap', 'moderate' or 'expensive'. I think this allows for a broader range of applications and I prefer it to just exploring costs for one or two (even if more aligned with the real-world).
> > >
> > > Finally, regarding the number of control sets, the reviewer raises a good point as the empirical evaluation does seem limited, as they always have 7. It would be good if this could be justified on some real-world application and to explore how performance changes over a variety of control sets.

---

### Official Review · Reviewer_mBSK · 2023-07-07

**Soundness:** 3 good
**Presentation:** 3 good
**Contribution:** 3 good
**Rating:** 7
**Confidence:** 3

**Summary:**

The authors study the problem of Bayesian optimization with cost-varying variable subsets (BOCVS) where in each iteration, the learner chooses a subset of query variables and specifies their values while the rest are randomly sampled. Each chosen subset has an associated cost. The authors analyze how the availability of cheaper control sets helps in exploration and reduces overall regret.

**Strengths:**

The paper is with good structures and is well-written. The problem is interesting and potentially has wide applications. The authors provide a profound theoretical analysis. The experimental results show the benefits of the proposed method.

**Weaknesses:**

Experimental results on a real-world application could make the paper stronger.



**Questions:**


1. How to choose the control set? How to choose their values?

2. The examples in the experiment section are based on simulators. A real-world example could strengthen the paper.

---

> ### Author Rebuttal · Authors · 2023-08-07
>
> Thank you for your time spent reading our paper and writing this review.
>
> **Questions**:
>
> 1. As part of the problem specification in Sec. 4, the learner is given a collection $\mathcal I \subseteq 2^{[d]}$ of control sets indexed by 1,2, ..., $m \coloneqq |\mathcal I|$. These are the control sets of which the learner is required to pick one at each iteration in the BO process. As described in Sec. 4.1, Algorithm 1's procedure to choose a control set and the partial query at each iteration is as follows:
>     1. Compute $g_t \coloneqq \max_{(i, \mathbf x^i)  \in [m]\times \mathcal X^i} \mathbb E[u_{t-1}([\mathbf x^i, \mathbf X^{-i}])]$, i.e., the expected UCB of the best control set and best partial query given that control set. Practically speaking, this expectation and all expectations are approximated to arbitrary accuracy using Monte Carlo sampling. All acquisition maximizations are performed with L-BFGS-B with random restarts as is standard for BO acquisition maximization.
>     2. Collect every control set $i$ that fulfills the condition $\max_{\mathbf x^i \in \mathcal X^i} \mathbb E[u_{t-1}([\mathbf x^i, \mathbf X^{-i}])] + \epsilon_t \geq g_t$ into a set $\mathcal S_1$. These are the control sets whose best expected UCB are $\epsilon_t$-close to the best value $g_t$.
>     3. Further reduce this set $\mathcal S_1$ to $\mathcal S_2$ by retaining only the control sets with the lowest cost. This step ensures that Algorithm 1 uses cheap control sets when available.
>     4. Finally, it chooses the control set from $\mathcal S_2$ with the largest expected UCB. This step ensures that, in the case of multiple control sets having the same cost after Step 3, Algorithm 1 chooses the 'best' among them. The partial query is then the one that maximizes the expected UCB given this control set.
>
> 2. We wish to point out that the plant and airfoil experiments are based on real-world datasets. The simulators are built on these real-world datasets in an attempt to be as close to a real-world experiment as reasonably possible. Unfortunately, we do not have the means (e.g., equipment, budget, domain experts, etc) to conduct real-world experiments. We hope you will understand the limitations we work with.
>
> We hope we have answered your questions satisfactorily. If you have any other questions that you wish for us to address, please let us know and we will be glad to respond further.

---

> > ### Comment · Reviewer_mBSK · 2023-08-21
> >
> > I thank the authors for the clarification, and my questions have been addressed.  I keep the score unchanged.

---

### Decision · Program_Chairs · 2023-09-21

**Decision:**

Accept (poster)

**Comment:**

This manuscript introduces an interesting twist on the types of problems typically considered in Bayesian optimization wherein the experimenter may choose to fix a chosen subset of the input variables -- for a cost that depends on the subset chosen -- with the remaining variables being sampled randomly. The authors provide a UCB-style algorithm for this setting and an accompanying regret bound.

The reviewers universally agreed that this setting and the presented algorithm and analysis were of interest to the NeurIPS community, and that the developments in this manuscript are correct and were presented clearly. On this basis, it would be a valuable addition to the conference program.

The response and discussion period proved useful in clarifying some issues, and I strongly recommend that the authors take this discussion into account while updating their manuscript.